



# The role of soil pH on soil carbonic anhydrase activity

Joana Sauze[1*], Samuel P. Jones[1], Lisa Wingate[1], Steven Wohl[1] and Jérôme Ogée[1*]

[1]INRA, UMR 1391 ISPA, F-33140 Villenave d'Ornon, France

[*]Author(s) for correspondence: J. Sauze (joana.sauze@ inra.fr) and J. Ogée (jerome.ogee@inra.fr)

**Abstract**. Carbonic anhydrases (CAs) are metalloenzymes present in plants and microorganisms that catalyse the interconversion of $CO_2$ and water to bicarbonate and protons. Because oxygen isotopes are also exchanged during this reaction, the presence of CA also modifies the contribution of soil and plant $CO^{18}O$ fluxes to the

global budget of atmospheric $CO^{18}O$. The oxygen isotope signatures ($\delta^{18}O$) of these fluxes differ as leaf water pools are usually more enriched than soil water pools, and this difference is used to partition the net $CO_2$ flux over land into soil respiration and plant photosynthesis. Nonetheless, the use of atmospheric $CO^{18}O$ as a tracer of land surface $CO_2$ fluxes requires a good knowledge of soil CA activity. Previous studies have shown that significant differences in soil CA activity are found in different biomes and seasons but our understanding of the

environmental and ecological drivers responsible for the spatial and temporal patterns observed in soil CA activity is still limited. One factor that has been overlooked so far is pH. Soil pH is known to strongly influence microbial community composition, richness and diversity in addition to governing the speciation of $CO_2$ between the different carbonate forms. In this study we investigated the $CO_2$-$H_2O$ isotopic exchange rate ($k_{iso}$) in six soils with pH varying from 4.5 to 8.5. We also artificially increased the soil CA concentration to test how pH

and other soil properties (texture and phosphate content) affected the relationship between $k_{iso}$ and CA concentration. We found that soil pH was the primary driver of $k_{iso}$ after CA addition and that the chemical composition (i.e. phosphate content) played only a secondary role. We also found an offset between the $\delta^{18}O$ of the water pool with which $CO_2$ equilibrates and total soil water (i.e. water extracted by vacuum distillation) that varied with soil texture. The reasons for this offset are still unknown.

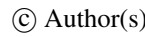


## 1 Introduction

The build up of carbon dioxide ($CO_2$) in the atmosphere is increasing rapidly because of anthropogenic activities (IPCC, 2013). The terrestrial biosphere currently absorbs and stores about 25% of anthropogenic $CO_2$ emissions as a result of a small disequilibrium whose amplitude reflects the land $CO_2$ sink strength and accounts for the

gross fluxes of $CO_2$ taken up by the terrestrial biosphere during photosynthesis and released back to the atmosphere during respiration (Le Quéré et al., 2015). It is clear from recent studies that this disequilibrium is highly variable from year to year with climate and is difficult to measure directly (Ballantyne et al., 2012; Gurney and Eckels, 2011; Poulter et al., 2014; Le Quéré et al., 2015). Currently this disequilibrium is estimated as a residual term in atmospheric budgets of $CO_2$ after reconciling the various fluxes between the oceans, the

atmosphere and anthropogenic emissions (including land use change). These mass budgets rely heavily on coupled climate-carbon models that require accurate representations of how key ecosystem processes such as respiration and stomatal conductance respond to changes in climate and other environmental factors (Friedlingstein et al., 2006). However, it is difficult to evaluate the performance of these models at large scales, as it is difficult to measure gross $CO_2$ fluxes directly (Wingate et al., 2009). Therefore, additional datasets and

tools that can track the behaviour of these processes and bring independent information on how to constrain their representation in models are now urgently required.

One potential approach advanced over recent years takes advantage of the observed variability in the oxygen isotope composition of $CO_2$ molecules in the atmosphere ($\delta^{18}O_a$) (Ciais et al., 1997; Cuntz, 2003; Farquhar et al., 1993; Francey and Tans, 1987; Welp et al., 2011; Wingate et al., 2009). This variability in $\delta^{18}O_a$ is driven

principally by the seasonal and inter-annual variability in the oxygen isotope composition of leaf and soil water pools that are strongly regulated by climate (Welp et al., 2011). Furthermore, large differences between the oxygen isotope composition of soil and leaf water pools exist and can be used to track rapidly the relative contributions of soil and leaf $CO_2$ exchange (Ciais et al., 1997; Farquhar et al., 1993; Francey and Tans, 1987; Welp et al., 2011; Wingate et al., 2010). This large-scale and rapid hydration of $CO_2$ by the biosphere is

accelerated by the family of carbonic anhydrase enzymes (CAs), that are ubiquitous in bacteria, algae, fungi and plants (Badger, 2003; Elleuche and Poggeler, 2010; Moroney et al., 2001; Smith and Ferry, 2000). In leaves the activity and concentration of CAs are high enough to expect that $CO_2$ diffusing out of the leaf is near full isotopic equilibrium with leaf water (Farquhar and Cernusak, 2012; Gillon and Yakir, 2001). In soils full isotopic equilibration between $CO_2$ and water can also occur below a certain depth (Miller et al., 1999; Tans,

1998) but will depend strongly on the distribution and activity of CA in the soil profile (Gangi et al., 2015; Wingate et al., 2009). This is because when the rate of $CO_2$ diffusion through a soil layer exceeds the CA-catalysed $CO_2$ hydration rate in that layer, full isotopic equilibration cannot occur (Tans, 1998; Wingate et al., 2008, 2009, 2010). Thus variations in soil CA activity dictate the shallowest depth where full isotopic equilibration between $CO_2$ and water can occur.

By compiling datasets of depth-resolved soil water $\delta^{18}O$ composition and soil-air $CO^{18}O$ exchange rates for a range of biomes, Wingate et al. (2009) found a tendency for larger soil CA activities in warmer and drier regions, and proposed 3 relatively crude but spatially-explicit scenarios of soil CA activity at the global scale (Wingate et al., 2009). Subsequently, using the lower range of soil CA activity estimates made by Wingate et al. (2009), an atmospheric $CO^{18}O$ inversion was performed and led to a surprisingly high rate of global

photosynthesis, of *ca*. 175 GtC yr⁻¹ over the period 1980-2010 (Welp et al., 2011). This global scale estimate of



photosynthesis over land was also highly sensitive to the range of soil CA activities used, demonstrating that a better understanding of the environmental and ecological drivers of soil CA activity was key to reduce the uncertainty in large scale gross $CO_2$ fluxes using atmospheric $CO^{18}O$ budgets.

Changes in the abundance and diversity of soil microbial communities were proposed as possible drivers of the observed spatial and temporal changes in soil CA activity (Seibt et al., 2006; Wingate et al., 2008, 2009, 2010). In particular, soil pH is known to strongly influence microbial community composition, richness and diversity (Fierer and Jackson, 2006; Griffiths et al., 2011; Hartman et al., 2008; Lauber et al., 2009) and could thus influence soil CA activity indirectly *via* changes in the microbial populations, with different CA requirements and isoforms. Soil pH should also influence CA-driven $CO_2$ hydration kinetics directly as CA reactivation is known to be limited by its de-protonation with a $pK_a$ around 7.2 (Rowlett et al., 2002). This may not be true for intra-cellular CA activity, as it has been shown that soil micro-organisms have the ability to regulate and maintain their intracellular pH within one pH unit near neutral (Krulwich et al., 2011). However, in certain micro-organisms, extracellular CAs have also been found (*e.g.* Hopkinson et al., 2013) whose activity should be directly affected by external (soil) pH. Thus a direct link between soil CA activity and soil pH should exist.

Actually, part of the reported variations in soil CA activity derived from the isotopic exchange rates between soil water and $CO_2$ can be explained by differences in soil pH. This is because soil CA activities are often reported relative to the un-catalysed $CO_2$-$H_2O$ isotopic exchange rate ($k_{iso,uncat}$), usually assumed equal to *ca.* 0.012 s$^{-1}$ at 25°C. However, because soil pH governs the speciation of $CO_2$ between the different carbonate forms, with dissolved $CO_2$ being predominant only in acidic environments (pH < 6), $k_{iso,uncat}$ is not the same for all soils and is strongly reduced in alkaline conditions (Mills and Urey, 1940; Uchikawa and Zeebe, 2012). Thus for the same soil CA activity – or more precisely for the same soil $CO_2$-water isotopic exchange rate ($k_{iso}$) – the enhancement factor relative to the *true* un-catalysed rate would actually be much greater in alkaline soils than for acidic ones.

The chemical composition of the soil is another potentially important factor that should be considered when reporting soil CA activity. Several studies have shown that some anions commonly found in soils could act as CA inhibitors or activators, depending on their ability to exchange protons. For example, at neutral pH, phosphate ions were reported to be activators of bovine α–CA as $CO_2$ hydration rates increased up to 6.5-fold relative to a solution without phosphates, whilst sulphate ions on the other hand were shown to act as a weak inhibitor on the same α–CA (Rowlett et al., 1991). The presence of these ions also modifies, sometimes dramatically, the pH response of CA activity *in vitro* (Rowlett et al., 1991), questioning our previous idea that pH might be the only chemical factor controlling soil CA activity.

The aim of this study was to investigate the relationship between soil CA activity and soil pH. For this we used a setup that allowed us to retrieve simultaneously the soil CA activity and the δ$^{18}$O of the soil water pool with which $CO_2$ equilibrates, without destructive sampling. Using six different soils differing in pH by almost 4 pH units, we investigated the influence of soil pH on the $CO_2$ hydration rate ($k_h$) and $CO_2$-$H_2O$ equilibration ($k_{iso}$). We also artificially increased the CA concentration in each soil by adding solutions of bovine CA. This CA isoform was chosen because it is well characterised in terms of enzymatic activity (Uchikawa and Zeebe, 2012) and pH response (Rowlett et al., 1991). Thus it was possible to investigate whether CA concentrations and soil pH were the only factors affecting the activity of this exogenous CA. Following the arguments elaborated above, we hypothesised that exogenous CA activity should be inhibited in acidic soils, but that the native soil phosphate concentration might also influence the activity of CA for soils differing in pH.



## 2 Material and methods

### 2.1 Soil sampling

A range of soils that differed naturally in terms of pH, texture, land use and chemical composition were investigated (Table 1). Soil samples from Le Bray, a maritime pine forest located at about 20km southwest of

Bordeaux (France), were collected in November 2014 (LeBray1) and April 2016 (LeBray2). The four other soils were sampled from croplands. The soils from Planguenoual (France, 95km northwest of Rennes) and Grignon-Folleville (France, 20km southwest of Paris) were collected in May 2013. More details about these soils can be found in Achat et al. (2014). The soil from Pierrelaye (France, 30km northwest of Paris) was sampled in October 2014 and May 2016 but mixed in one batch and the soil from Toulouse was sampled once in May 2016.

All soil samples were taken from the soil surface (0-15cm) after removal of the coarse litter elements. They were sieved with a 4mm mesh, homogenised and air-dried for several weeks in the laboratory. A first set of experiments was conducted in March 2015 with the soils from LeBray1, Planguenoual and Grignon-Folleville. Prior to gas exchange measurements, 330 to 440 g of soil were re-packed in a 500-mL Teflon pot to obtain a common soil depth (*ca*. 7 cm). For each soil type, we prepared three re-packed Teflon pots. Each soil pot was

then irrigated with 125 mL of water and allowed to drain for 12 to 24h at room temperature and light conditions. Drainage was facilitated by a small hole (3mm inner diameter) drilled at the bottom of the pot to avoid the accumulation of water and anoxic conditions in deeper soil layers. During gas exchange measurements this hole was closed with a Teflon screw. Preliminary results from this first set of experiments indicated substantial evaporation-induced isotopic enrichment of soil water in the top layer, as well as soil-to-soil variations in water-

filled pore space (WFPS), that could complicate the interpretation of the results. Thus a second set of experiments was conducted in June 2016 where the WFPS was better controlled and soil evaporation was minimised by allowing the soils to drain prior to measurement inside a dark chamber controlled at 21°C and with a saturating water vapour generated from an evaporating reservoir filled with the same water that was used for irrigation. The soils from LeBray2, Pierrelaye and Toulouse were chosen for this second set of experiments, to

minimise differences between soil texture, whilst keeping a large range of pH (Table 1). Prior to gas exchange measurements, 280 to 300 g of soil were re-packed in a 500-mL Teflon pot to a common soil depth of *ca*. 5-6 cm. Each soil pot was then irrigated in order to reach a WFPS of 25% and left in the dark chamber for 24h. All Teflon pots used in the experiment had a constant surface area of 41.85 cm$^2$.

### 2.2 Carbonic anhydrase addition

For a set of gas exchange measurements, lyophilised $\alpha$-CA powder from bovine erythrocytes (C3934-100MG, Sigma-Aldrich, France) was diluted into the irrigation water. For each set of experiments, CA concentrations of *ca*. 24 and 80 mg L$^{-1}$ were used. Apart from this addition of CA into the irrigation water, all other steps were kept identical and as described for the measurement soils without CA addition.

### 2.3 Experimental setup and working sequence

Prior to gas exchange measurements, each soil pot was sealed with a screw-tight lid and used as a flow-through gas exchange chamber. An acclimation time of at least 20 minutes was used to allow the soil column to re-equilibrate to the new air supply $CO_2$ composition. The soil $CO_2$ efflux and its oxygen isotopic composition were then measured using the experimental setup illustrated in Fig. 1. To simultaneously retrieve soil CA activity, reported here as the $CO_2$-$H_2O$ isotopic exchange rate $k_{iso}$, and the $\delta^{18}O$ of the soil water pools with which $CO_2$





equilibrates ($\delta_{sw-eq}$), we designed a system that allowed us to measure $CO_2$ isotope fluxes under two, quasi-simultaneous isotopic steady states that only differ in the isotopic composition of the $CO_2$ entering the soil chamber (Fig. 1). The air supplied to the chamber came directly from a compressed air tank during the first steady state (SS1) and from a mix between dry, $CO_2$-free air (from a synthetic air generator) and a tank of pure

$CO_2$ during the second steady state (SS2). During SS2 mixing valves adjusted the $CO_2$ concentration of the inlet air to maintain it close to the value of the inlet air used in SS1 within acceptable error (423 ± 5 ppm), whilst their oxygen isotope compositions differed markedly (Fig. 2). The transition between SS1 and SS2 was operated by means of a three-way valve (Fig. 1) and a transition period of 20 minutes was necessary to attain the new steady state (Fig. 2). A full sequence of measurements lasted about 1h (Fig. 2) and consisted of 2 steady states. During

each steady state, the by-pass (*i.e.* the air entering the soil chamber) and the outlet of the chamber were alternately selected *via* a manifold connected to a stable isotope $CO_2$ analyser (Fig. 2). During the first set of experiments each working sequence was repeated three times on each soil and CA treatment (pseudo-replication) at a temperature of 25°C. During the second set of experiments each soil and CA treatment were made in triplicates but measured only once over a single working sequence (true replication) at a temperature of 21°C.

To account for possible non-linearity and drift of the stable isotope analyser during the experiments, gas from two calibration tanks of known $CO_2$ concentration and isotopic composition were regularly recorded in-between sample measurements (Fig. 2). For both sets of experiments the calibration tanks, whose $^{12}C^{16}O_2$ and $^{12}C^{16}O^{18}O$ mixing ratios bracketed our measurements, were measured at approximately 16 min intervals, consistent with the expected stability of the analyser (see below).

### 2.4 $CO_2$ mixing ratio and stable isotope measurements

Mixing ratios of $^{12}C^{16}O_2$, $^{13}C^{16}O_2$ and $^{12}C^{16}O^{18}O$ were measured using an Isotope Ratio Infrared Spectrometer (IRIS, Delta Ray, Thermo Fisher Scientific, USA). The cell pressure was controlled and maintained at 100 mbar throughout the experiment. The air sample passed through a multi-pass Herriot cell at a flow of 85 mL min$^{-1}$ with a total path length of 5 m, leading to a theoretical residence time in the analyser of *ca*. 35 seconds. To

minimise carry-over effects caused by this residence time, each line (inlet or chamber air or calibration tanks) were measured for 2 minutes and only the last 40 s of measurements were averaged to provide a single mean and standard deviation.

Calibration tank mixing ratios for the different isotopologues ($^{12}CO_2$, $^{13}CO_2$ and $CO^{18}O$) were averaged as described above, and interpolated in time using a spline function. These interpolated time-series were then used

to perform a two-point calibration regression on the mixing ratios for each sample measurement. Total $CO_2$ mixing ratio was computed following Wingate et al. (2010):

$$[CO_2]=\frac{[^{12}C^{16}O^{16}O]+[^{13}C^{16}O^{16}O]+[^{12}C^{16}O^{18}O]}{0.999179} \qquad (1)$$

where the factor 0.999179 accounts for the presence of $^{12}C^{16}O^{17}O$ in the gas mixture. The $\delta^{18}O$ of $CO_2$ was expressed on the VPDB-$CO_2$ scale using the formula:

$$\delta^{18}O\text{-}CO_2=0.5\frac{[^{12}C^{16}O^{18}O]/[^{12}C^{16}O_2]}{0.00208835}-1 \qquad (2)$$

where 0.0020835 is the $^{18}O/^{16}O$ isotope ratio of the VPDB-$CO_2$ reference standard (Allison et al., 1995) and the factor 0.5 accounts for the fact that there is two oxygen atoms per molecule of $CO_2$ but only one $^{18}O$ atom in $^{12}C^{16}O^{18}O$.





Standard deviations on $CO_2$ mixing ratios of the different isotopologues were propagated using Eq. 1 to provide a measurement error on total $CO_2$. In contrast measurement error on the isotope ratios were not calculated using Eq. 2 but computed from the standard deviation over the last 20 s of measurements of the instantaneous ratio $[^{12}C^{16}O^{18}O]/[^{12}C^{16}O_2]$. This is because fluctuations in one $CO_2$ isotopologue was always highly correlated with

fluctuations in the other $CO_2$ isotopologue leading to much smaller fluctuations in their ratios than the one calculated from simple error propagation using Eq. 2. Using this approach, typical errors on $[CO_2]$ and $\delta$ $^{18}O$-$CO_2$ values were around 0.1 ppm and 0.3‰, respectively.

Under steady state conditions (*i.e.* during SS1 or SS2 in Fig. 2), and according to the mass balance of total $CO_2$ in the chamber headspace, the soil $CO_2$ efflux is proportional to the total $CO_2$ concentration difference between

the inlet and outlet airstreams:

$$F = \frac{u_{in}}{S}\left(c_{out} - c_{in}\right) \tag{3}$$

where $u_{in}$ (mol s$^{-1}$) is the flow rate of dry air on the inlet of the chamber, $S$ is the soil surface (m$^2$) and $c_{in}$ and $c_{out}$ are the mixing ratios of total $CO_2$ (mol mol$^{-1}$) in the air entering and leaving the chamber respectively. Because these mixing ratios were determined on a dry air basis (because of the Nafion dryer upstream before the $CO_2$

isotope analyser) only the flow of dry air on the inlet of the chamber was required to perform the mass balance. The fluxes of $^{12}C^{16}O_2$ ($^{16}F$) and $^{12}CO^{18}O$ ($^{18}F$) can be computed using Eq. 3, and the oxygen isotopic signature of the soil $CO_2$ flux ($\delta_F = 0.5^{18}F/^{16}F/0.00208835 - 1$, also expressed on the VPDB-$CO_2$ scale) can thus be calculated from the $^{12}C^{16}O_2$ concentrations and $\delta^{18}O$ of the inlet ($^{16}C_{in}$, $^{18}\delta_{in}$) and outlet ($^{16}C_{out}$, $^{18}\delta_{out}$) air:

$$\delta_F = \frac{^{16}c_{out}\,^{18}\delta_{out} - ^{16}c_{in}\,^{18}\delta_{in}}{^{16}c_{out} - ^{16}c_{in}} \tag{4}$$

For each steady state, 3 or 4 inlet/outlet measurements were performed leading to 3 or 4 individual values of $\delta_F$ from which a mean and standard deviation could be computed.

### 2.5 Theoretical retrieval of soil CA activity and $\delta_{eq}$

Assuming uniform soil properties (*i.e.* uniform soil porosity, moisture and temperature), $\delta_F$ can be computed as (Tans, 1998; Wingate et al., 2010):

$$\delta_F = \delta_{eq} + \varepsilon_D + \frac{V_{inv}C_a}{F}\left(\delta_{eq} - \delta_a\right) \tag{5}$$

where $\delta_{eq}$ (‰ VPDB-$CO_2$) is the $CO_2$ oxygen isotopic composition in equilibrium with soil water, $\varepsilon_D$ (-8.7‰) is the oxygen isotope fractionation factor during diffusion of $CO_2$ in air, $C_a$ (mol m$^{-3}$) and $\delta_a$ (‰VPDB-$CO_2$) are the concentration and $\delta^{18}O$ of $CO_2$ in the air at the soil-air interface, respectively, $F$ is the soil $CO_2$ efflux ($\mu$mol m$^{-2}$ s$^{-1}$) and $V_{inv}$ (m s$^{-1}$) is the so-called piston velocity. In the following we will assume full mixing inside

the chamber so that $\delta_a = ^{18}\delta_{out}$ and $C_a = c_{out}p/8.3144/T$ where $p$ (Pa) and $T$ (K) are air pressure and temperature inside the chamber headspace. The piston velocity is a function of soil moisture and temperature and soil CA activity only (Tans, 1998; Wingate et al., 2010) so that it should be the same between the two steady states. Because $c_{out}$ and $F$ were also maintained constant between the two steady states it was possible to retrieve $\delta_{eq}$ and $V_{inv}$ from the two steady-state measurements:

$$\delta_{eq} = \frac{-\varepsilon_D\left(\delta_{a,1} - \delta_{a,2}\right) + \delta_{F,2}\delta_{a,1} - \delta_{F,1}\delta_{a,2}}{\delta_{a,1} - \delta_{a,2} + \delta_{F,2} - \delta_{F,1}} \tag{6a}$$



$$V_{inv} = \frac{F}{C_a} \frac{\delta_{F,2} - \delta_{F,1}}{\delta_{a,1} - \delta_{a,2}} \qquad (6b)$$

where $\delta_{a,1}$ and $\delta_{a,2}$ are $\delta_a$ during steady states 1 and 2 and $\delta_{F,1}$ and $\delta_{F,2}$ are the corresponding $\delta_F$, computed from Eq. 4.

Strictly speaking Eq. 5 is valid only for a semi-infinite soil column. In our experiments the soil depths were of a

few centimetres only and mass transport was not possible at the bottom of the soil column (*i.e.*, zero $CO_2$ flux), because the microcosms were closed at the bottom. With this new boundary condition, Eq. 5 should be slightly modified (see Appendix A for a full derivation):

$$\delta_F = \delta_{eq} + \tilde{\varepsilon}_D + \frac{\tilde{V}_{inv} C_a}{F}\left(\delta_{eq} - \delta_a\right) \qquad (7)$$

with $\tilde{\varepsilon}_D = \varepsilon_D\left(1 - z_1/z_{max} \tanh\left(z_{max}/z_1\right)\right)$ and $\tilde{V}_{inv} = V_{inv} \tanh\left(z_{max}/z_1\right)$, where $z_1 = D_{iso}/V_{inv}$ and $z_{max}$ is soil depth.

The right-hand side of Eq. 6b was then used to estimate $\tilde{V}_{inv}$ and $V_{inv}$ was solved iteratively to satisfy the equation $\tilde{V}_{inv} = V_{inv} \tanh\left(V_{inv} z_{max}/D_{iso}\right)$, from which $z_1$ and then $\tilde{\varepsilon}_D$ and $\delta_{eq}$ could be deduced (using Eq. 6b replacing $\varepsilon_D$ by $\tilde{\varepsilon}_D$).

The soil $CO_2$–$H_2O$ isotopic exchange rate ($k_{iso}$, in s$^{-1}$) was then derived from the piston velocity according to:

$$k_{iso} = \frac{V_{inv}^2}{D_{iso} B \theta} \qquad (8)$$

where $B$ (m$^3$ m$^{-3}$) is the solubility coefficient for $CO_2$ in water (Weiss, 1974), $\theta$ (m$^3$ m$^{-3}$) is the volumetric soil water content, $D_{iso} = D_{eff} / (1 - \varepsilon_D)$ and $D_{eff}$ (m$^2$ s$^{-1}$) is the effective diffusivity of gaseous $CO_2$ through the soil matrix (Tans, 1998; Wingate et al., 2010). The latter was computed using the formulation of Moldrup et al. (2003) for repacked soils: $D_{eff} = (\phi - \theta)^{2.5}/\phi\, D_0$, where $\phi$ (m$^3$ m$^{-3}$) is total soil porosity and $D_0$ (m$^2$ s$^{-1}$) is the molecular diffusivity of $CO_2$ in soil air at temperature $T_s$ (K): $D_0 = 1.381\ 10^{-5}\ (T_s/273.15)^{1.81}$ (Massman, 1998).

The soil $CO_2$–$H_2O$ isotopic exchange rate $k_{iso}$ was further converted into a $CO_2$ hydration rate ($k_h$). Following Uchikawa and Zeebe (2012) we have:

$$k_h = 2k_{iso}\left\{1 + \frac{C}{S} - \sqrt{1 + \frac{2}{3}\frac{C}{S} + \left(\frac{C}{S}\right)^2}\right\}^{-1} \qquad (9)$$

where $C$ (mol m$^{-3}$) is the $CO_2$ concentration in soil water and $S = [H_2CO_3] + [HCO_3^-] + [CO_3^{2-}]$. Assuming that the ratio $C/S$ is close to its equilibrium value (this assumption is actually required to derive Eq. 9), the ratio $k_h/k_{iso}$

is only a function of temperature and pH (Uchikawa and Zeebe, 2012). In acidic soils, this ratio approaches 3 at any temperature, because there are three oxygen atoms in the $CO_2$-$H_2O$ system and in this pH range, $CO_2$ is the dominant dissolved inorganic carbon species.

Following the same reasoning as in Ogée et al. (2016) for OCS hydrolysis, the soil $CO_2$ hydration rate can also be expressed as a function of bulk CA concentration [CA] (mol m$^{-3}$):

$$k_h = k_{h,uncat}(T, pH) + \frac{k_{cat}}{K_m}(T, pH)\left[CA\right] \qquad (10)$$

where $k_{h,uncat}$ (s$^{-1}$) is the un-catalysed $CO_2$ hydration rate at a given temperature $T$ (K) and pH and $k_{cat}$ and $K_M$ are the (spatially-averaged) CA-catalysed maximum hydration rate and Michaelis-Menten constant at the same





temperature and pH. The expected pH dependency of $k_h$ and $k_{iso}$ for different levels of CA concentrations are shown in Fig. 3.

Values of $\delta_{eq}$ were converted into a soil water isotope composition equivalent ($\delta_{sw\text{-}eq}$, in ‰VSMOW) according to (Brenninkmeijer et al., 1983): $\delta_{sw\text{-}eq} = \delta_{eq} + 0.20(T_s - 297.15)$. According to Wingate et al. (2009) this $\delta_{sw\text{-}eq}$

should correspond to the soil water $\delta^{18}O$ at a depth $z_{eq}$ (m) given by:

$$z_{eq} \approx 2\sqrt{2\ln 2}\, z_l \qquad\qquad (11)$$

**2.6 Water extraction and isotopic measurements**

These estimated profiles of soil water $\delta^{18}O$ were further compared to $\delta^{18}O$ measurements of soil water extracts ($\delta_{sw}$). For this, after completion of the full gas exchange sequence shown in Fig. 2, soil samples were collected at

1, 2 and 4 cm below the soil surface and stored in Weaton glass jars with parafilm in a fridge. Water from these soil samples was then extracted by vacuum distillation and the extracted water analysed for stable isotope composition using a Triple Isotope Water Analyser (TIWA 45EP; Los Gatos Research Inc., CA, USA) coupled to a liquid auto sampler (PAL System, Switzerland). The $\delta^{18}O$ values of soil water samples were calibrated on the VSMOW-SLAP scale using three internal laboratory water standards that covered the expected range of $\delta^{18}O$

in soil water (-10.16 ± 0.06 ‰, -5.59 ± 0.14 ‰ and +5.21 ± 0.13 ‰ in 2015 and -10.31 ± 0.06 ‰, -4.84 ± 0.06 ‰ and +0.62 ± 0.06 ‰ in 2016, on the VSMOW-SLAP scale). Two internal standards (the most depleted and more enriched ones) were used for calibration whilst the third internal standard was used for quality check. These in-house standards were kept in 25L kegs that were over-pressured with dry air and measured against IAEA standards before and after the experiments, with no drift observed.

Both soil water samples and internal working standards were transferred into 2mL glass vials and the vials were then closed with pre-pierced PTFE caps and silicone septa. Vials with internal standard waters were interspaced every five sample vials following the International Atomic Energy Agency (IAEA) recommendations. A small water volume (0.2-1.0 µL) from each vial was sampled using a 5-µL syringe (SGE Analytical Science, Ringwood, Australia) and injected through a septum in a vaporiser unit maintained at 80°C to help vaporise the

water under vacuum immediately upon injection. The vapour was then transferred through a Teflon tube to the pre-evacuated optical cavity of the water isotope analyser. Before each measurement the syringe was rinsed three times in deionised water. Each vial was then measured eight times in total and only the last five measurements, subject to filtering, were retained and averaged. The accuracy (the mean absolute difference between calibrated and true $\delta^{18}O$ values) and reproducibility (the standard deviation of these means) on the $\delta^{18}O$ measurements of

this internal standard used for quality check were always below 0.15 ‰ and 0.1 ‰ respectively.

**2.7 Phosphate concentration measurements**

Because phosphate ions can act as either strong CA activators (Rowlett et al., 1991) or CA inhibitors (Rusconi et al., 2004), total phosphate concentration in the different soils was also measured using the water extraction and colorimetric method (Van Veldhoven and Mannaerts, 1987). On 10 g of dry soil we added 99 mL of deionised

water and 1 ml of a biocide (Toluene) to stop any microbial activity. Soil suspensions were incubated at 20°C for 16 h on an agitating roller, then sampled with plastic syringes and filtered through 0.2 mm membrane filters. The filtered solutions were then analysed for phosphate concentrations (mg(P) L$^{-1}$) using a malachite green





colorimetric method (Van Veldhoven and Mannaerts, 1987). Results were then expressed on a dry soil mass basis $(mg(P) \; kg(soil)^{-1})$.

## 3 Results

### 3.1 Illustration of the non destructive soil CA activity measurement method

From each sequence and steady state, it was possible to compute a relationship between the soil $CO_2$–$H_2O$ isotopic exchange rate, $k_{iso}$ and the isotope composition of soil water in equilibration with soil $CO_2$, $\delta_{sw\text{-}eq}$ by combining Eqs. 7 and 8. This approach, when presented graphically, leads to a plot of up to six curves (3 for each steady state, see Fig. 4) that intersect at a near-common place within the $k_{iso}$-$\delta_{sw\text{-}eq}$ space. Combining the two steady states from the same sequence and using the iterative procedure described above, it is also possible to

estimate $k_{iso}$ and $\delta_{sw\text{-}eq}$ separately, as demonstrated by the symbols in Fig. 4. These values corresponded closely to the intersection points of the two curves for each steady state in the $k_{iso}$-$\delta_{sw\text{-}eq}$ space (Fig. 4). Errors on the $CO_2$ isotope measurements were also algebraically propagated into the equations in order to estimate uncertainties on $k_{iso}$ and $\delta_{sw\text{-}eq}$. The repeatability of the measurements between the three sequences was very good with a standard deviation equal to or lower than the propagated error on individual estimates (i.e., the spread of the squares in

Fig. 4 was always smaller than the error bars on each individual square). Sometimes the intersection between the two lines was not as clearly defined as the one shown in Fig. 4 but the combination of the two steady states always provided very consistent and repeatable estimates of both $k_{iso}$ and $\delta_{sw\text{-}eq}$ between the different sequences. For example, in the experiment shown in Fig. 4, we obtained $k_{iso}$ values of $0.022 \pm 0.005 \; s^{-1}$, $0.025 \pm 0.006 \; s^{-1}$ and $0.025 \pm 0.005 \; s^{-1}$ and $\delta_{sw\text{-}eq}$ values of -11.3 ± 0.6, -11.5 ± 0.7 and -11.2 ± 0.3 ‰VSMOW for the three

sequences. These estimated values of $\delta_{sw\text{-}eq}$ were in this case depleted compared to the $\delta^{18}O$ of irrigation water (-10.1 ‰VSMOW) and that of cryogenically extracted soil water at the equilibration depth $z_{eq}$ (Fig. 5). Similar results were also observed on LeBray2 where the water pool "seen" by $CO_2$ had an isotopic composition ($\delta_{sw\text{-}eq}$, black circles in Fig. 5) that was strongly depleted (by about 5‰) compared to the cryogenically-extracted soil water pool (blue squares in Fig. 5). In contrast, more enriched $CO_2$-derived $\delta_{sw\text{-}eq}$ values and shallower $z_{eq}$ were

found in soils containing a larger clay fraction (i.e. Planguenoual and Folleville, see Table 1), also in much better agreement with the $\delta^{18}O$ profile of cryogenically-extracted soil water (Fig. 5).

### 3.2 Effect of soil pH on soil CA activity

The native (*i.e.* without any addition of exogenous $\alpha$-CA during irrigation) isotopic exchange rates ($k_{iso,native}$) of the six soils were always higher than the un-catalysed rate ($k_{iso,uncat}$) and tended to increase slightly with more

alkaline conditions (Fig. 6). These values of native isotopic exchange rates are consistent with what we would theoretically predict using $\beta$-CA concentrations between 10 and 80 mg $L^{-1}$ (Fig. 3).

The addition of exogenous CA generally led to higher $k_{iso}$ values compared to the native rates, and also enhanced $CO_2$ hydration rates $k_h$, with marked differences depending on the pH range (Fig. 6). On the most acidic soils, the addition of exogenous $\alpha$-CA barely increased $k_h$ above its native rate ($k_{h,native}$), by 0.1 $s^{-1}$ or less (the native

rate was around 0.06 $s^{-1}$), but within the uncertainties on the measurements. On the other hand for the most alkaline soils (Toulouse, Folleville) $k_h$ increased to about 20 $s^{-1}$ with 24 mg $L^{-1}$ of CA added to the irrigation



water and up to 65-100 s$^{-1}$ at 80 mg L$^{-1}$. Results from the soils with more neutral pH (Planguenoual, Pierrelaye) were intermediate between these two cases with enhanced hydration rates of the order of 10 s$^{-1}$ or less.

This influence of soil pH on the enhancement of $k_h$ by exogenous CA was anticipated as the $k_{cat}/K_M$ value of α-CA is known to be strongly reduced in acidic pH with a pH response of the form (Rowlett et al., 1991):

$$\frac{k_{cat}}{K_m} = \left(\frac{k_{cat}}{K_m}\right)_{max} \frac{1}{1+10^{pK_a-pH}} \qquad (12)$$

To test whether our results only reflected the pH response of the exogenous α-CA, we rewrote Eq. 10 as follows:

$$k_h = k_{h,native} + \frac{k_{cat}}{K_m}[CA]_{exogenous} \qquad (13)$$

where $k_{h,native}$ (s$^{-1}$) represents the native value of $k_h$ and $[CA]_{exogenous}$ (mol m$^{-3}$) is the concentration of exogenous CA in soil water. For a given pH (and temperature) the difference $\Delta k_h = k_h - k_{h,native}$ should then be proportional to $[CA]_{exogenous}$ and the slope of the relationship should be given by $k_{cat}/K_M$ and thus be influenced by soil pH. The theoretical pH response of $\Delta k_h$ at the two CA concentration values used in this study (24 and 80mg L$^{-1}$) is shown in Fig. 6b, using Eq. 12 with $pK_a = 7.1 \pm 0.5$ and $(k_{cat}/K_M)_{max} = 30 \pm 7$ s$^{-1}$ μM$^{-1}$ and a molar mass of 30 kg mol$^{-1}$, typical values for bovine α-CA (Lindskog and Coleman, 1973; Rowlett et al., 1991; Uchikawa and Zeebe, 2012). For LeBray1, Folleville and Toulouse, our results were in very close agreement with Eq. 12 for the two different CA concentrations we tested, but this was not the case for the other soils. For LeBray2 and Pierrelaye, the observed enhanced hydration rates were smaller than the ones predicted by Eq. 12 while for Planguenoual, they were higher.

## 4. Discussion

### 4.1 Can we predict the enhancement in soil CA activity with exogenous CA?

Results presented in Fig. 6b demonstrate that a low (acidic) soil pH clearly inhibits the non-native, additional hydration rate of $CO_2$ induced by a supply of exogenous CA to the soil water. Our data from three of the soils (LeBray1, Folleville and Toulouse) agreed remarkably well with the pH response described by Eq. 12 and parameterised with $k_{cat}/K_M$ and $pK_a$ values previously estimated from independent studies on the same α–CA than the one used here (Uchikawa and Zeebe, 2012) or other bovine CA (Rowlett et al., 1991). This indicates that our gas exchange method to estimate $CO_2$ hydration rates in soil water is robust, despite possible complications caused by $CO_2$ diffusion through the soil matrix and the potential for heterogeneity in soil water content and pore space in our microcosms. A further possible complication could have arisen because of the necessity to subtract the native hydration rate from our $\Delta k_h$ calculations. This approach could have introduced a possible bias in our calculations of $\Delta k_h$ if the native hydration rates were markedly different between soils with and without CA addition, i.e., if the addition of water with exogenous CA over the 12h-24h prior to our gas exchange measurements was enough to induce changes in microbial growth and diversity compared to what we would expect in soils where only water was added. To rule out this possibility we estimated the bacterial and fungal abundance using qPCR for some of our microcosms and could not find any clear trend in the number of 16S and 18S gene copies with the amount of exogenous CA added to the soil (not shown). These results, although only preliminary, suggest that within the timeframe of our experiment, exogenous CA addition should



not affect the native $CO_2$ hydration rates or community structure and thus our $\Delta k_h$ estimations should not be biased.

The observed values of $\Delta k_h$ were not always consistent with those predicted for three of the soils (LeBray2, Pierrelaye and Planguenoual). One possible reason for these discrepancies could be that the model we are using to derive $k_{iso}$ and thus $k_h$ from our gas exchange data (Eq. 8) assumes that the soil column is homogeneous in terms of soil water content, temperature, porosity, CA concentration and respiration rate (Tans, 1998, see also Appendix A). Care was taken to remain as close as possible to these conditions: the soils had been sieved and homogenised before being placed into the soil chambers, the irrigation of the soil was performed at least 12h prior to the gas exchange measurements and the soil microcosms were immersed in a water bath to minimise temperature gradients during the gas exchange measurements. Furthermore, in 2016 we also increased the preparation time to 24h and minimised soil water evaporation and isotopic enrichment (see Material and Methods). However, despite these precautions, soil water content and its oxygen isotope composition was not always homogeneous throughout the soil column (Fig. 5).

Also, on the most alkaline soils, we noticed that the $CO_2$ mixing ratio on the outlet of the soil microcosm was not always constant but decreased slightly, indicating that steady state was not reached. This could be explained by the fact that these alkaline soils contain a large pool of total dissolved inorganic carbon that takes much longer to re-equilibrate after a change in the $CO_2$ concentration in the microcosm headspace, especially if this concentration differed markedly from the $CO_2$ concentration seen by the soil prior to measurement. On these soils, the acclimation time of 20 minutes was certainly too short but was chosen as a compromise in order to minimise other possible artefacts caused by soil evaporation whilst the microcosm was flushed with dry air during the measurements.

In order to explore the possible consequences of non-steadiness and soil water inhomogeneity on our $k_{iso}$ estimates, we also used a numerical model that simulates explicitly the transport and rate of change of the different $CO_2$ isotopologues throughout the soil column and inside the chamber headspace. The model was similar to the one used in Gangi et al. (2015) but with prescribed vertical profiles of soil water content ($\theta$) and isotopic composition ($\delta_{sw}$). The model was ran over the entire sequence shown in Fig. 2 and three model parameters were optimised in order to find the best match between the modelled and observed time-series of $CO_2$ mixing ratio and its carbon and oxygen isotopic composition in the chamber headspace. These model parameters were the ratio $k_{iso}/k_{iso,uncat}$ (assumed constant through the soil column), the $CO_2$ mixing ratio of the air prior to connecting the microcosm to the air supply and a possible offset between $\delta_{sw}$ and $\delta_{sw,eq}$ (also assumed constant throughout the soil column). The latter parameter seemed necessary given the results shown in Fig. 5. Soil $CO_2$ production rate was assumed to be uniform throughout the soil column and computed iteratively to match the observed $CO_2$ efflux. Soil temperature was set to the constant value of the water bath and vertical profiles of soil water content and isotopic composition ($\delta_{sw}$) were prescribed from depth-resolved measurements (Fig. 5). Surprisingly, the results from this numerical model differed only marginally from those shown in Fig. 5 and Fig. 6 (not shown). Values of $\Delta k_h$ were slightly affected by non-steady-state effects, either positively (Pierrelaye) or negatively (Planguenoual). Soil water inhomogeneity could also affect $\Delta k_h$ values slightly both positively (Folleville) or negatively (LeBray1). Overall the discrepancies between $\Delta k_h$ estimates and the theoretical predictions (Eq. 12) were only marginally reduced, even after non-steadiness and soil water inhomogeneity had been accounted for.





Another factor that could explain the deviation of $\Delta k_h$ from theory is the presence of phosphate ions in the soil solution that could either activate or inhibit CA compared to its activity in the absence of such anions (Rowlett et al., 1991; Rusconi et al., 2004). We tested this hypothesis by exploring how the ratio between $\Delta k_h$ predicted by Eq. 12 ($\Delta k_{h,theory}$) and the observed $\Delta k_h$ varied with total phosphate concentration ($P_i$), as well as with the

concentrations in mono- and di-hydrogen phosphate ions ($HPO_4^{2-}$ and $H_2PO_4^-$ respectively). Although the relationships between $\Delta k_{h,theory}/\Delta k_h$ and the different phosphate ion concentrations were quite dispersed, we could observe a positive trend (not shown). Also two of the soils with the highest total $P_i$ and $H_2PO_4^-$ molar concentrations (LeBray2 and Pierrelaye) had also the largest $\Delta k_{h,theory}/\Delta k_h$ ratio, corresponding to an inhibitory factor of about 10 in Pierrelaye and even higher in LeBray2. This could indicate that phosphate ions act as an

inhibitor of the exogenous CA used in our experiments, explaining the reduced response to CA addition in these two soils (Fig. 6).

**4.2 With which soil water pool does the $CO_2$ equilibrate?**

Our results also revealed large differences between the isotopic composition of the water pool "seen" by the $CO_2$ ($\delta_{sw-eq}$) and that of cryogenically extracted soil water ($\delta_{sw}$), with more depleted $\delta_{sw-eq}$ values compared to $\delta_{sw}$

(Fig. 5). Interestingly very similar "offsets" between $\delta_{sw}$ and $\delta_{sw,eq}$ were also predicted by the numerical model (not shown), except for LeBray1 where even larger offsets were found. For a given soil the offset did not seem to vary with soil CA activity (i.e. the difference between $\delta_{sw}$ and $\delta_{sw,eq}$ was the same for soils with and without CA addition) and, at least for the only soil tested, did not seem to be affected by soil water content (similar offsets were observed between LeBray1 and LeBray2). However, in-between the different soils, it seemed that

those with the highest CA activity (Planguenoual, Folleville) also had the smallest offset, close to zero.

The exact reason for this offset between $\delta_{sw}$ and $\delta_{sw,eq}$ is still unknown. Noting that $\delta_{sw}$ and $\delta_{sw,eq}$ are estimated from measurements coming from different analysers, we verified that the calibrations of the two analysers were consistent with one another. We thus pressurised pure $CO_2$ into a keg partially full of water of known isotopic composition and let the water-$CO_2$ mixture equilibrate for several weeks. The pure $CO_2$ was then diluted into

$CO_2$-free air to reach ambient $CO_2$ concentrations and the air mixture was analysed with our $CO_2$ isotope analyser. We found a small difference of about -0.31‰ between the $\delta^{18}O$ of the equilibrated $CO_2$ and the $\delta^{18}O$ of the water in the keg. Clearly, such a bias would only explain a small fraction of the measured offset between $\delta_{sw}$ and $\delta_{sw,eq}$, down to -6‰ on some soils. Also the fact that this offset cancels in soils with high CA activity indicates that our calibration scheme is clearly not the only cause of the existence of such an offset.

A possible explanation for the observed difference between $\delta_{sw}$ and $\delta_{sw,eq}$ could be that, at any given depth, soil water is not isotopically homogeneous and that $CO_2$ "sees" a different water pool to that extracted during cryogenic distillation, with different thermodynamic and chemical properties between the different soil water pools. This idea has been proposed by several studies already. For example Hsieh et al. (1998) allowed pure $CO_2$ to equilibrate for several weeks with different soils at different water contents and found that the isotopic

composition of equilibrated $CO_2$ could differ by several ‰ compared to the $\delta^{18}O$ of the soil water extracted by vacuum distillation, even at relatively high (i.e. 32%) gravimetric water contents. They explained this difference by recognising that soil surfaces contain a lot of ions that could modify the isotopic composition of the "bound" water pool and also the $CO_2$-$H_2O$ isotopic fractionation factor.





More recently, Chen et al. (2016) performed laboratory experiments that suggest the existence of two isotopically distinct pools of water around hydrophilic materials such as silage, litter or soil organic matter. They found a negative apparent isotopic fractionation between total water (extracted by cryogenic distillation) and unconfined water (estimated by water liquid-vapour equilibration), suggesting a depletion of the water bound to the hydrophilic material. They also found that the magnitude of this apparent fractionation increased with the solid to water ratio. To reconcile these results with ours, we would need to assume that $CO_2$ equilibrates with bound water, even when exogenous CA is added to the soil. This is somewhat surprising, because once in solution we would expect the exogenous CA to be equally spread between bound and unbound water. Another explanation could be that water around the CA reaction sites is depleted. Chen et al. (2016) found large apparent fractionation factors with water adsorbed onto casein, another protein found in milk. However according to their theory, at high water contents (or low solid-to-water ratios), the fractionation factor should vanish. In addition Uchikawa and Zeebe (2012) found that the isotopic equilibration between $BaCO_3$ and water was not affected by the presence of CA in the solution, thus rejecting the hypothesis of different water composition around the CA reaction sites. Clearly, more experiments on $CO_2$-$H_2O$ equilibration in soils such as those performed by Hsieh et al. (1998) are needed to better understand the underlying mechanisms leading to this apparent oxygen isotope disequilibrium between soil $CO_2$ and soil water, even below the equilibrium depth.

## 5. Conclusion

Our experimental results demonstrate that our two steady-state approach is robust and sensitive enough to detect changes in the $CO_2$-$H_2O$ isotope exchange rate when the concentration of CA enzyme in the soil matrix is augmented artificially. We also found that natural variations in soil pH had a strong control over the variability of soil CA activity, with a smaller influence of the phosphate ion concentration, and these variations reassuringly followed similar patterns to those observed in other studies on α-CA activity in buffered solutions. However, although α-CAs may be present in certain soil microbial communities with a high abundance of phototrophs such as cyanobacteria and micro-algae, the majority of microbial CAs in soils are more likely represented by the β-CA class (Smith and Ferry, 2000). In addition, β-CAs are seldom active externally like α-CAs and are rather found in the internal cell components of the microbe, in particular the cytoplasm (e.g. Merlin et al., 2003). Thus, although β-CAs also exhibit a strong dependence of CA activity with pH (Rowlett et al., 2002), it remains to be investigated whether the location and relative abundance of different CAs in soil communities modifies the expected relationship with pH. In addition it is not clear whether the impact of anions such as phosphate ions will remain important when the CA is active internally. This was beyond the scope of the present study but is an obvious next step to be addressed in future experiments to help understand and model better the spatio-temporal variations in atmospheric $CO^{18}O$ at large scales.



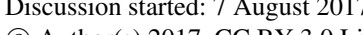


**Appendix A: derivation of Eq. 7 in the main text**

Following Tans (1998) we will assume that the $CO_2$ concentration profile within the soil column is driven by two processes: respiration, characterised by a production density $S$ (mol m$^{-3}$ s$^{-1}$) and diffusion, characterised by an effective diffusivity $D_{eff}$ (m$^2$ s$^{-1}$). At steady state, the mass balance equation thus writes (see also Eq. 3 in Tans, 1998):

$$D_{eff}\frac{d^2C}{dz^2} + S = 0 \,, \tag{A1}$$

where $C$ (mol m$^{-3}$) is the $CO_2$ concentration at depth $z$ (m) within the soil column. Assuming $S$ constant throughout the soil column (a fair assumption when working on repacked, temperature-controlled soil columns), and with the boundary conditions $C = C_a$ at $z = 0$ and $dC/dz = 0$ at $z = z_{max}$, the solution of Eq. A1 is (see for example Eq. 23a in Tans, 1998):

$$C(z) = C_a + \frac{S}{D_{eff}}\left[ zz_{max} - \frac{z^2}{2} \right] = 0 \,. \tag{A2}$$

Denoting by $R$, $R_S$ and $R_{eq}$ the $^{18}O/^{16}O$ ratio of soil air $CO_2$, soil respired $CO_2$ and $CO_2$ in equilibrium with soil water, respectively, the steady-state $CO^{18}O$ mass balance equation is (see also Eq. 9 in Tans, 1998):

$$D_{iso}\frac{d^2RC}{dz^2} - B\theta k_{iso}C(R - R_{eq}) + SR_S = 0 \,. \tag{A3}$$

Defining $y = RC$ Eq. A3 becomes:

$$z_1^2\frac{d^2y}{dz^2} - y = -y_S - R_{eq}C(z) \text{ with } z_1^2 = \frac{D_{iso}}{B\theta k_{iso}} \text{ and } y_S = \frac{R_S S}{B\theta k_{iso}} \,. \tag{A4}$$

The general solution of this differential equation is of the form: $y(z) = Ae^{-z/z_1} + Be^{+z/z_1} + Y(z)$ where $A$ and $B$ are constants to be defined and $Y$ is a particular solution of Eq. A4. Choosing $Y$ of the form $Y = az^2 + bz + c$, the coefficients $a$, $b$ and $c$ must satisfy Eq. A4 for any depth $z$. Using the expression of $C(z)$ from Eq. A2, this gives:

$$Y(z) = -R_{eq}\frac{S}{2D_{eff}}z^2 + R_{eq}\frac{S}{D_{eff}}z_{max}z + R_{eq}C_a + y_S - R_{eq}\frac{S}{D_{eff}}z_1^2 \,. \tag{A5}$$

With the boundary conditions $y = C_a R_a$ at $z = 0$ and $dy/dz = 0$ at $z = z_{max}$, the solution of Eq. A4 can be found (i.e. constants $A$ and $B$ can be identified) and this gives:

$$y(z) = \left[ C_a(R_a - R_{eq}) + R_{eq}\frac{Sz_1^2}{D_{eff}} - y_S \right]\frac{e^{-z/z_1} + \xi^2 e^{+z/z_1}}{1+\xi^2} + Y(z) \,, \text{ with } \xi = e^{-z_{max}/z_1} \,. \tag{A6}$$

The $CO_2$ and $CO^{18}O$ fluxes at the soil surface are given by:

$$F = D_{eff}\frac{dC}{dz}\bigg|_{z=0} \text{ and } F_{iso} = D_{iso}\frac{dy}{dz}\bigg|_{z=0} \,. \tag{A7}$$

From Eq. A2 we get $F = Sz_{max}$ and from Eq. A6 we obtain:

$$F_{iso} = \left[ V_{inv}C_a(R_{eq} - R_a) - \frac{z_1}{z_{max}}F(\alpha_D R_{eq} - R_S) \right]\frac{1-\xi^2}{1+\xi^2} + \alpha_D R_{eq}F \,, \tag{A8}$$

where $\alpha_D = D_{iso}/D_{eff}$. Defining $R_F = F_{iso}/F$ and using the delta notation (i.e., $\delta = R/R_{std} - 1$ where $R_{std}$ is the $^{18}O/^{16}O$ ratio of the international standard VPDB$_g$), Eq. A8 becomes:

$$\delta_F = \left[ \frac{V_{inv}C_a}{F}(\delta_{eq} - \delta_a) - \frac{z_1}{z_{max}}(\delta_{eq} + \varepsilon_D - \delta_S) \right]\frac{1-\xi^2}{1+\xi^2} + \delta_{eq} + \varepsilon_D \,, \tag{A9}$$



where $\varepsilon_D = \alpha_D - 1$ and noting that the second-order term $\varepsilon_D \delta_{eq}$ has been discarded. Now assuming $R_S = R_{eq}$ (or equivalently $\delta_S = \delta_{eq}$) Eq. A9 simplifies to Eq. 8 in the main text.



**Authors contribution**

JS, SJ, LW, SW and JO conceived and designed the experiment. JS, SJ and SW conducted the gas-exchange and water isotope measurements. JS, SJ and JO analysed the data. JS, JO and LW wrote the manuscript. All authors commented and contributed to the final version.





**Acknowledgements**

Pierre-Alain Maron and Virginie Nowak are gratefully acknowledged for quantitative PCR analysis performed on some of our soil samples. We also thank Alain Mollier for helping us with the phosphate concentration measurements. This project has received funding from the European Research Council (ERC) starting grant SOLCA under the European Union's Seventh Framework Programme (FP7/2007-2013) (Grant Agreement No. 338264), the French Agence National de la Recherche (ANR) (Grant Agreement No. ANR-13-BS06- 0005-01) and the Institut National de la  Recherche Agronomique (INRA) departments EFPA and EA (PhD studentship of JS).





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



**Tables and Figures**

| | Le Bray1 | Le Bray2 | Planguenoual | Pierrelaye | Grignon-Folleville | Toulouse |
|---|---|---|---|---|---|---|
| **Land use** | pine plantation | pine plantation | cropland | cropland | cropland | cropland |
| **Coordinates** | 44°42'N 0°46'W | 44°42'N 0°46'W | 48°32'N 02°34'W | 49°02'N 02°13'E | 48°50'N 01°56'E | 43°32'N 01°30'E |
| **pH** | 4.1 (*4.1*) | 4.8 (*4.1*) | 6.3 (*6.3*) | 7.6 (*7.8*) | 8.2 (*8.1*) | 8.5 (*8.5*) |
| **Sand content %** | *94.7* | *94.7* | *43.7* | *82.2* | *11.0* | *43.8* |
| **Silt content %** | *2.6* | *2.6* | *41.5* | *8.7* | *60.3* | *38.2* |
| **Clay content %** | *2.7* | *2.7* | *14.8* | *9.1* | *28.7* | *18* |
| **Total N (g kg$^{-1}$)** | *31.2* | *31.2* | *16.6* | *11.5* | *14.3* | *7.5* |
| **Total C (g kg$^{-1}$)** | *1.2* | *1.2* | *1.6* | *0.83* | *1.2* | *0.59* |
| **Phosphates (mg kg$^{-1}$)** | 4.85 | 6.93 | 2.88 (*3.0*) | 13.6 | 0.53 (*0.5*) | 1.4 |

Table 1: main characteristics of the soils investigated in this study. Numbers in italics indicate literature data.





Figure 1: Schematic of the experimental setup used to estimate simultaneously the $CO_2$-$H_2O$ isotope exchange rate ($k_{iso}$) in a soil microcosm and the oxygen isotopic composition of the soil water pool with which the $CO_2$ equilibrates ($\delta_{sw\text{-}eq}$). The soil microcosm is thermally regulated using a water bath and flushed with $CO_2$ in dry air whose oxygen isotopic composition was either enriched (-3.8‰ $VPDB_g$) or depleted (between -24‰

5   and -27‰ $VPDB_g$, depending on the experiment).

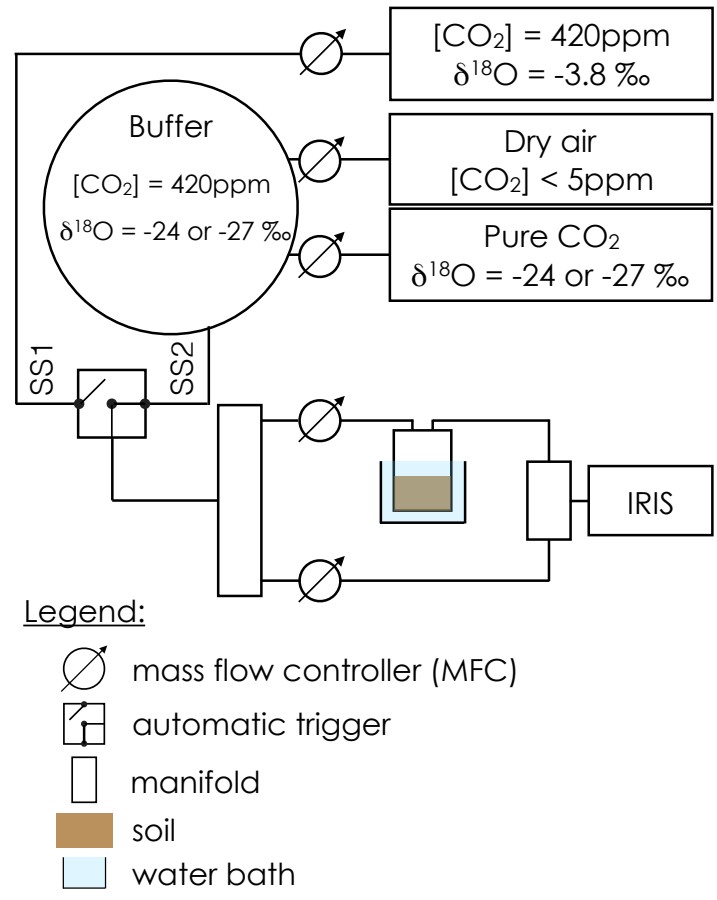





Figure 2: Typical time-series of the measured $CO_2$ mixing ratio and isotope composition ($\delta^{18}O$) over the course of a working sequence. The sequence is composed of 7 steps: (1) two calibration bottles spanning the expected range of $CO_2$ mixing ratios, (2) inlet and outlet lines of the soil microcosm, measured 4 times consecutively, using a $CO_2$ with an enriched $\delta^{18}O$ (steady state 1), (3) calibration bottles, (4) the outlet of the chamber during

5    the switch of the air supplying the soil chamber (front), (5) calibration bottles, (6) inlet and outlet lines of the soil chamber, measured 4 times consecutively, using a $CO_2$ with a depleted $\delta^{18}O$ (steady state 2) and (7) calibration bottles.

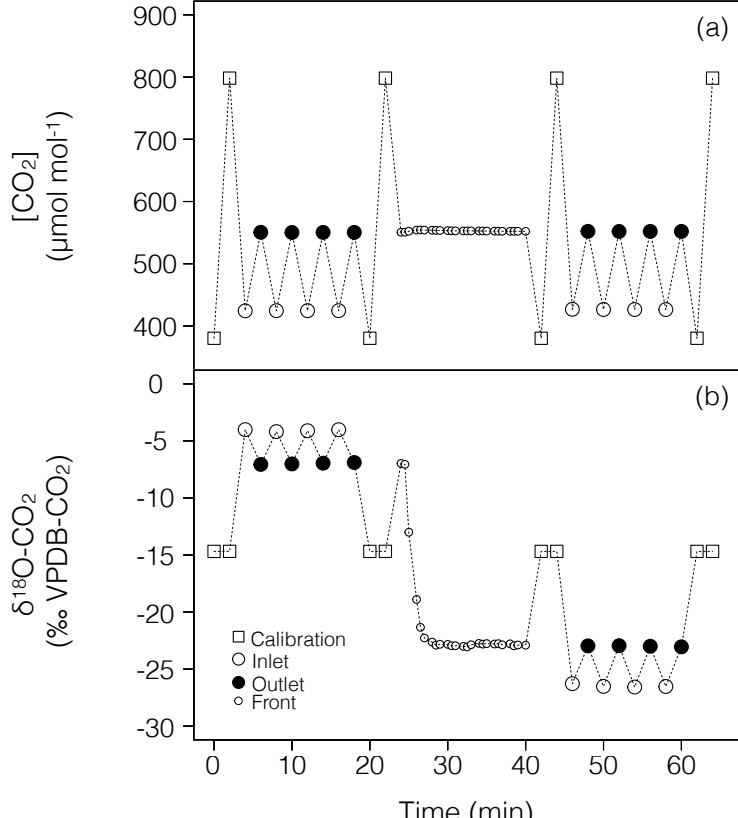



Figure 3: Theoretical rates of $CO_2$ hydration ($k_h$) and $CO_2$-$H_2O$ oxygen isotope exchange ($k_{iso}$) as a function of pH, for 3 levels of carbonic anhydrase concentration. These theoretical curves have been obtained using enzymatic parameters of $k_{cat}/K_M = 70$ s$^{-1}$ μM$^{-1}$ and $pK_a = 7$, which are typical of βCAs, the most abundant CA isoform expected in soils.

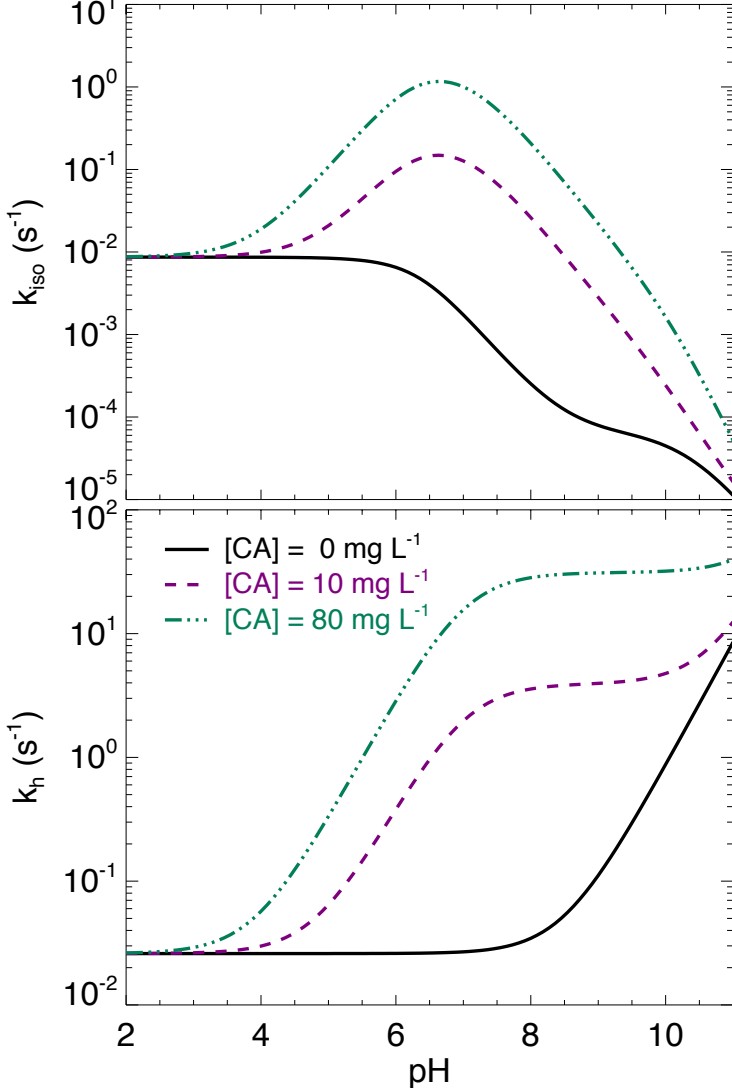





Figure 4: The $CO_2$-$H_2O$ isotopic exchange rate ($k_{iso}$) and isotopic composition of soil water equilibrated with $CO_2$ ($\delta_{sw}$) retrieved using the two-steady-state approach described in the main text, for LeBray1 soil and an $\alpha$CA addition of 24 mg $L^{-1}$. Relationships between $k_{iso}$ and $\delta_{sw}$ for steady-state 1 (dotted lines) and steady-state 2 (solid lines) are also shown. In this example 3 sequences were used, resulting in 3 curves for each steady state and 3 intersection points. The pH-dependent, un-catalysed $CO_2$-$H_2O$ isotopic exchange rate is also indicated by the grey horizontal line.

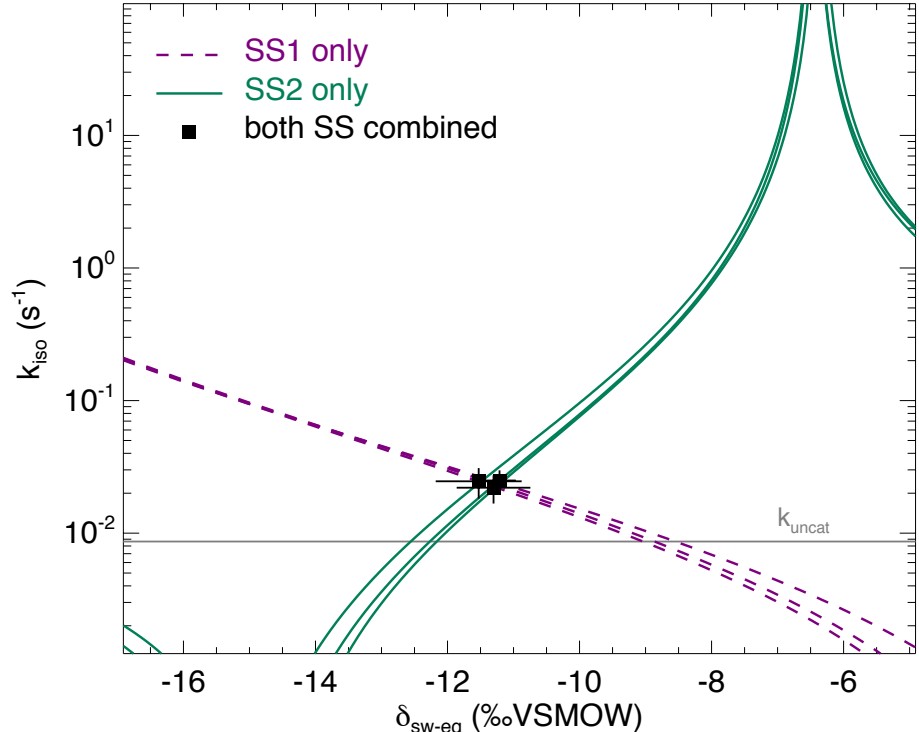



Figure 5: The isotopic composition of soil water at different depths in the replicated soil microcosms from each site, estimated either by vacuum distillation and water isotope analysis (blue squares) or online $CO_2$-$H_2O$ isotopic exchange using the two steady-state approach (black circles, see text). Profiles for the different CA treatments are plotted together without distinction. The blue vertical line also indicates the isotopic composition of the irrigation water used for the re-wetting of the air-dried soils.

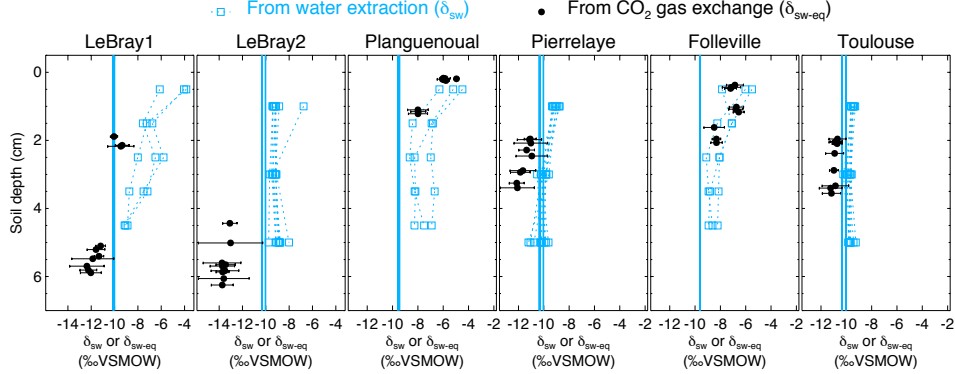





Figure 6: The measured $CO_2$-$H_2O$ isotopic exchange rates ($k_{iso}$) in the different soils for different levels of α-CA addition (top) and associated enhancement hydration rates ($k_h$ - $k_{h,native}$) caused by the α-CA addition (bottom). In the top panel, the un-catalysed isotope exchange rate ($k_{iso,uncat}$) is shown for reference (black dotted curve). Native rate (grey curve in top panel) and theoretical rates above the native rate (green and purple curves) are also shown, using $k_{cat}/K_M = 30 \pm 5$ s$^{-1}$ μM$^{-1}$ and $pK_a = 7.1 \pm 0.5$.

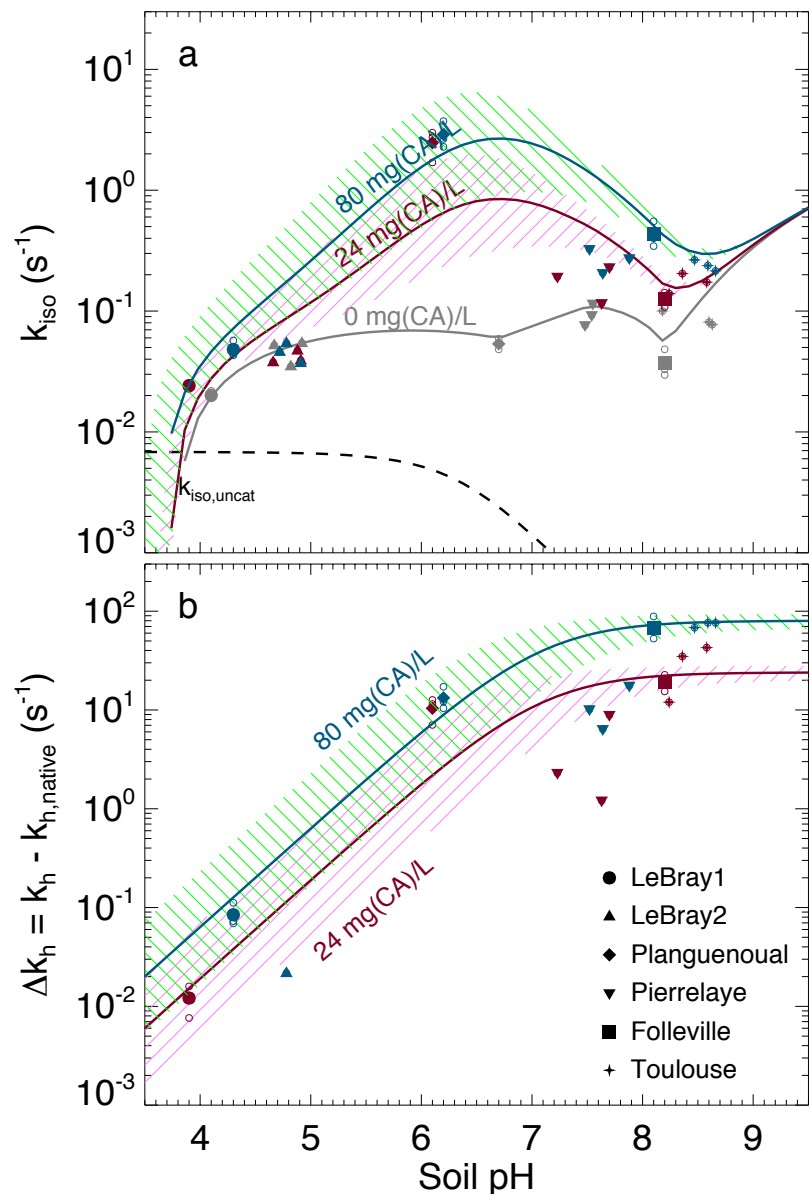