# Peer review of "The role of soil pH on soil carbonic anhydrase activity"

_Biogeosciences, 2017_

## Referee Comment (RC1) · Anonymous Referee #3 · 5 Oct 2017

Importance:

The stable isotope of oxygen (d18O), is routinely used to characterize ecosystem processes from the cell to the globe (Werner et al., 2011, biogeosciences). The enzymatic activity of carbonic anhydrase (CA) enhances the exchange of the source water d18O signal with CO2 emitted from points of respiration. We can better constrain estimates of ecosystem metabolism by increasing our knowledge of CA activity under different environmental conditions. Sauze et al. set out to fill these gaps related to soil in our understanding through a series of innovative experiments and modeling. They quantified the isotopic exchange rate ($k_{iso}$) between H2O and CO2 in soils with a range of pH.

The authors devised small incubation units to which they could add water of known isotopic composition, known CO2 gas, and varying amounts of α-CA (28 and 80 mg $L^{-1}$). By controlling the environment, mainly through maintaining a near constant relative humidity, the authors were able to isolate the impact of soil pH on CA as determined by $k_{iso}$.

They found that the experimental results from three of the soils were congruent with their modeling. For the other three soils that did not conform to the modeling prediction, the authors infer that heterogeneity from the soil column, non steady-state conditions, or phosphate ions may have led to the aberrations.

The experiment was carried out well and the analyses were quite thorough, although a few statistical analyses with regards to differences in the isotopic composition of the source water and extracted water are needed. The manuscript needs a better description of the experiments performed and the hypotheses/questions asked of each steady-state iteration. I suggest a simple table or figure will suffice. This will let the reader keep better track of 1) some of the terminology used to describe different parameters (e.g., native, non-native), 2) sample size, and the results in general. In the introduction the real-world differences between α and β CA should be discussed. Furthermore, the use of α-CA can be defended more strongly upfront rather than toward the conclusion of the manuscript. I also am surprised by the differences in d18O between the soil extracted and equilibrated water values. This is important evidence in the ongoing debate of the two water world hypotheses (Kirchner, James W. "A double paradox in catchment hydrology and geochemistry." *Hydrological Processes* 17, no. 4 (2003): 871-874.; McDonnell, J. J. (2014), The two water worlds hypothesis: ecohydrological separation of water between streams and trees?. WIREs Water, 1: 323–329.; Sprenger, M., H. Leistert, K. Gimbel, and M. Weiler (2016), Illuminating hydrological processes at the soil-vegetation-atmosphere interface with water stable isotopes, Rev. Geophys., 54, 674–704.Vargas, A. I., Schaffer, B., Yuhong, L. and Sternberg, L. d. S. L. (2017), Testing plant use of mobile vs immobile soil water sources using stable isotope experiments. New Phytologist, 215: 582–594.).

Detailed comments:

Line 12: delete "nonetheless"

Line 3: Biosphere "absorbs" - unconventional way of describing flux and pools
Line 17: delete "advanced over recent years" given the literature list dating back to 1997 (and earlier?), the method is established.
Line 37- perhaps simple instead of crude.
Line 40- since you introduce these values can you report the accepted global photosynthesis rates?

Page3

Line 4: You introduce microbial communities as a possible explanation, but not how the microbes might alter CA activity. I assume it might have something to do with total biomass and functional characteristics of the communities.
Line 18: please provide a citation for this value.
Line 23- I think you mean soil solution here.

Perhaps specify in your hypotheses that you are primarily interested in the direct effects of pH and not the indirect effects anticipated from shifts in microbial diversity and function.

Line 30: I suggest to introduce the "native" or control term here and please explain the situation to which you arrived at estimates of "un-catalyzed rates" (page 9 line 28).
Line 32: Why were these concentrations chosen?
Lines32-33: Please edit this sentence, especially "the measurement soils without".
Line 38: Introduce the water bath here and how its temperature was maintained.

Line 3: The description of the two gases here is lacking. Is there not also CO2 in the compressed air-tank? It is not clear at this point why you have the two different gas sources. Please explain how you achieved different d18O compositions in the two gas sources.
Line 24: each is singular in this case: "Each line was measured", "only the last 40s of measurement was averaged"
Line 29- do you mean over the measurement time? i.e., the measurement period?

Line 1: what do you mean precisely by "were propagated"?
Line 17. The equation in parentheses is difficult to decipher, perhaps separate it from the text.
Line 25: Can you better define piston velocity here? It looks like eqn. 5 is a formulation of Fick's diffusion implementing Henry's law. In this case, is the piston velocity expressing a minimal exchange of gas at the soil-water boundary during equilibrium? Tans (1998) also discusses piston velocity within this context.
Line 29: delete "so-called"

Line 11: $D_{iso}$ is not defined here
Line 28: add carbonyl sulfide along with OCS

Line 10: refrigerator instead of "fridge"
Line 25: please edit" "to help vaporise the water under vacuum immediately upon injection", this does not read well.
Line 28: I assume this is a data filter and not a physical filter used in the analysis.
Line 29: "of the measurements"; please edit this whole sentence.

Line 8: what is "near-common"?
Line8: perhaps, inform the reader before the calculation is described that it is desirable to verify the $k_{iso}$ and $d_{sw-eq}$ independently.
Line9: what is "sequence" referring to here?
Line 20: the difference is roughly 1 per mil, can you check that the difference is significant?
Line 28: why native? Is this the same as the control?
Line 29: do they mention "un-catalyzed" before?

Equation 12: let the reader know that $(k_{cat}/K_m)max$ is explained previously in equation 10.

Line 33: I suggest to separate Fig. 6a and 6b. Is the apparent peak and subsequent decline of $K_{iso}$ explained in the results or the discussion?

Line 19: This is a little confusing, it can be read as th ough your are using exogenous CA as a tool to predict the enhancement in soil CA activity, or it can be read as a general question asking if we can predict the enhancement in soil CA activity when additional CA is introduced.

Line 20: Since this is the first sentence of your discussion, perhaps you can set the reader up for what the topic of discussion is for this paragraph. The term native doesn't appear until page 9 line 28 and I don't think the term non-native is ever defined in the prior text. This exemplifies why a table or figure explaining the experiment will help the reader.

How is the fact that three of your soils did not conform to your model reconciled within fig 6?

Line11: this sentence is a little convoluted, it reads at first as if the soil pH is going to have a response when in fact this metric is intrinsic to the soil.
Line 22: deviations from non-steady state instead of non-steadiness.
Line 26: "was run" or simply "ran"
Line 28: please explain "native hydration". Is this potential water remaining within the soil from the field or elsewhere that has potentially mixed with the irrigation water?

The check for non-steady state was cursory and not all the assumptions were easily understood. While the exercise is interesting, it is not possible for the reader to determine how robust the results are. I think the authors need to decide how important this issue is to their results and either fully address the issue to the best extent possible or shorten the explanation and report that the non-steady state effect needs to be addressed by further experimentation and modeling.

Conclusion- Can you bring out the larger relevance once more? How might the effect of pH influence our estimates of ecosystem carbon balance at different scales?

Figure 2. Please insert the step number within the top or bottom panel. Check the grammar within the caption, for example, I believe you want to say you measured the two calibration bottles in step 1.

Figure 3. I recommend that the use of a-CA in the experiments is clarified in the figure. Perhaps also how this figure is related to the experimental results.

Figure 4. It is probably worthwhile to report the results for all the soils or at least place them in the supplementary. Please reference where the $K_{uncat}$ is presented in the text.

Figure 6. The caption does not reference a or b. In the text, I could not find a reference to a. Please explain what appears to be model uncertainty.

---

## Referee Comment (RC2) · Anonymous Referee #2 · 10 Oct 2017

The paper "The role of soil pH on soil carbonic anhydrase activity" by Sauze, Jones, Wingate, Wohl, and Ogée explores the role of pH on soil carbonic anhydrase (CA) activity by combining a novel experimental setup with a rigorous model framework. The authors are thereby able to compare measured rates of oxygen isotope exchange and CO2 hydration, and their response to pH, versus theoretical expectations. The results of this study confirm in many cases the mechanistic understanding of the role of pH on soil CA activity. In the process, the authors reveal the potential role of of soil complexity on the bulk behavior of soils including heterogeneous distributions of water content, temperature, porosity, enzymes concentrations, and respiration rates. Using their model framework, Sauze et al. are able to evaluate some of these sources of variability and inform critical and current discussions in soil science, such as whether distinct isotopic pools of water exist in soils. This manuscript thus makes important contributions to the study of the role of soil CA activity and its pH dependence and to a

broader body of research in soil science.

P2L3: This sentence describing the role of the terrestrial biosphere in compensating for anthropogenic CO2 emissions is difficult to understand, and should be clarified.

P2L14: Is this the correct reference for direct CO2 measurements?

P2L33: Variations in soil properties affecting diffusion rates would also be important, and could be mentioned.

P3L14: More fitting would be to suggest that a direct link between at the activity of at least some CA in soils and soil pH should exist because the case was just made that the intracellular CA may not experience environmental pH fluctuations.

P3L22: This is an important point regarding the mode that CA enhancement has been reported in the past. The point would be more effective by clarifying the sentence more. For example, the 'enhancement factor' is not defined before its first mention in line 21 making it difficult for the reader to know how it is different from the uncatalyzed rate mentioned.

P7L32: What is the meaning of spatially-averaged here? Does this just mean that the kinetic parameters are average values for the volume or mass of soil, or should spatially-averaged refer to something more specific? If so, would be good to clarify.

P10L34: 16S and 18S rRNA or rDNA gene copies. No detectable difference in these gene copies does show no signifiant change in community structure in response to CA addition, but it does not necessarily mean that native CO2 hydration rates were unchanged because microbial communities may have modulated their CA gene expression and enzyme production rates, and thus native CO2 hydration rates, in response to the availability and activity of exogenous alpha CA.

P11L36: Results for these model results should be given, even if just summarized briefly as a % change from steady-state conditions. Also for P12L16. Could a figure for just one site be added to illustrate the difference between steady- and non steady-

state?

P12L1: Reader should be pointed to Table 1 to look for phosphate concentrations.

P13L1-16: Interesting results and informative discussion

Table 1: citations for literature data should be provided

Figure 1: the 'automatic trigger' terminology seems a bit odd if the text just calls the component a 3-way valve

Figure 3: What is the basis for the expectation that beta CAs are the most abundant in soils? Provide citation or justification.

Figure 4: It is not clear which lines and points in Figure 4 correspond with LeBray1 soil versus an $\alpha$CA addition of 24 mg L-1, which are both stated in the caption. If the $\alpha$CA data were plotted for some soils, wouldn't the kiso values be different? If they are not significantly different, as suggested in Fig 6, a justification for plotting results from the no-addition and addition should be given because that reasoning is not clear at the beginning of the results section. Would it be worthwhile switching the order of 3.1 and 3.2 or referencing 3.2 as justification?

Figure 5: May be useful to state why plotted without distinction (CA conc shouldn't affect result for water isotopic composition) and rest why CO2 gas exchange results shift with depth (Eq xx)

Figure 6: difficult to see diamond points - shift CA concentration labels. Why are some LeBray2 points missing in 6b? What are the open circles representing? State in caption.

---

## Author Comment (AC1) · 20 Nov 2017

**Response to Anonymous Referee #2[*]**

[*]Extracts from reviewer's original comments are indicated in *blue italic*
[*]Extracts from our original manuscript are indicated in *black italic*
[*]Proposed modification on our original manuscript are indicated in ***black bold italic***

*The paper "The role of soil pH on soil carbonic anhydrase activity" by Sauze, Jones, Wingate, Wohl, and Ogée explores the role of pH on soil carbonic anhydrase (CA) activity by combining a novel experimental setup with a rigorous model framework. The authors are thereby able to compare measured rates of oxygen isotope exchange and $CO_2$ hydration, and their response to pH, versus theoretical expectations. The results of this study confirm in many cases the mechanistic understanding of the role of pH on soil CA activity. In the process, the authors reveal the potential role of soil complexity on the bulk behaviour of soils including heterogeneous distributions of water content, temperature, porosity, enzymes concentrations, and respiration rates. Using their model framework, Sauze et al. are able to evaluate some of these sources of variability and inform critical and current discussions in soil science, such as whether distinct isotopic pools of water exist in soils. This manuscript thus makes important contributions to the study of the role of soil CA activity and its pH dependence and to a broader body of research in soil science.*

We are pleased that referee #2 appreciated the originality and significance of our study.

*P2L3: This sentence describing the role of the terrestrial biosphere in compensating for anthropogenic $CO_2$ emissions is difficult to understand, and should be clarified.*

We agree that the sentence was a bit long and we simplified and shortened it:

*The terrestrial biosphere currently **mitigates** about 25% of anthropogenic $CO_2$ emissions as a result of a small disequilibrium **between two large gross $CO_2$ fluxes, photosynthetic $CO_2$ uptake and respiratory $CO_2$ release** (Le Quéré et al., 2015).*

*P2L14: Is this the correct reference for direct $CO_2$ measurements?*

We meant "estimate" gross $CO_2$ fluxes, as they currently cannot be "measured" at scales above the organ or plot level. We changed the sentence and also added two extra references:

*(…) as it is difficult to **estimate** gross $CO_2$ fluxes directly **(Beer et al., 2010;** Wingate et al., 2009**, 2010**).*

*P2L33: Variations in soil properties affecting diffusion rates would also be important, and could be mentioned.*

We added this idea:

*Thus variations in soil CA activity **and $CO_2$ diffusion rates** dictate the shallowest depth*

*where full isotopic equilibration between $CO_2$ and water can occur.*

*P3L14: More fitting would be to suggest that a direct link between the activity of at least some CA in soils and soil pH should exist because the case was just made that the intracellular CA may not experience environmental pH fluctuations.*

We modified the sentence accordingly:

*Thus a direct link between (**at least a fraction of)** soil CA activity and soil pH should exist.*

*P3L22: This is an important point regarding the mode that CA enhancement has been reported in the past. The point would be more effective by clarifying the sentence more. For example, the 'enhancement factor' is not defined before its first mention in line 21 making it difficult for the reader to know how it is different from the uncatalyzed rate mentioned.*

We tried to clarify the sentence by explaining a bit more how the enhancement factor was defined previously and how we propose it should be defined from now on:

*This is because soil CA activities are often reported **as an enhancement factor** relative to **an un-catalysed $CO_2$-$H_2O$ isotopic exchange rate**, assumed equal to ca. 0.012 $s^{-1}$ at 25°C (**Miller et al., 1999**). However, because soil pH governs the speciation of $CO_2$ between the different carbonate forms, with dissolved $CO_2$ being predominant only in acidic environments (pH < 6), **the true un-catalysed rate** ($k_{iso,uncat}$) is not the same for all soils and is strongly reduced in alkaline conditions (Mills and Urey, 1940; Uchikawa and Zeebe, 2012). Thus for the same soil CA activity – or more precisely for the same soil $CO_2$-water isotopic exchange rate ($k_{iso}$) – the enhancement factor **should rather be defined** relative to the true un-catalysed rate ($k_{iso}/k_{iso,uncat}$) **and** would **then** be much greater in alkaline soils than **in** acidic ones.*

*P7L32: What is the meaning of spatially-averaged here? Does this just mean that the kinetic parameters are average values for the volume or mass of soil, or should spatially averaged refer to something more specific? If so, would be good to clarify.*

We replaced the term "spatially-averaged" by "community-averaged" to be more specific about the type of averaging.

*P10L34: 16S and 18S rRNA or rDNA gene copies. No detectable difference in these gene copies does show no significant change in community structure in response to CA addition, but it does not necessarily mean that native $CO_2$ hydration rates were un-changed because microbial communities may have modulated their CA gene expression and enzyme production rates, and thus native $CO_2$ hydration rates, in response to the availability and activity of exogenous alpha CA.*

We analysed rDNA gene copies, not rRNA, and this is now clarified in the text. We agree that an unchanged community structure does not necessarily translate into no change in CA

activity in response to exogenous CA addition. We thus introduced this possible caveat into our discussion and proposed it as a possible explanation of the reported discrepancies between observed and predicted $\Delta k_h$:

*This approach could have introduced a possible bias in our calculations of $\Delta k_h$ if the native hydration rates were markedly different between soils with and without CA addition, i.e., if the addition of water with exogenous CA over the 12h-24h prior to our gas exchange measurements was enough to induce changes in microbial growth and diversity **and/or their CA gene expression** compared to soils where only water was added. We estimated the bacterial and fungal abundance using qPCR for some of our microcosms and could not find any clear trend in the number of 16S and 18S **DNA** gene copies with the amount of exogenous CA added to the soil (not shown). These results suggest that, within the timeframe of our experiment, exogenous CA addition did not affect the community structure. **However, conservation of the community structure does not necessarily translate into conservation of the native $CO_2$ hydration rate as microbial communities may have modulated their CA gene expression in response to the availability and activity of exogenous CA. Actually, the observed values of $\Delta k_h$ were not always consistent with those predicted for three of the soils (LeBray2, Pierrelaye and Planguenoual), which may indicate changes in native $CO_2$ hydration rates with exogenous CA addition, that would have biased our $\Delta k_h$ estimations. Another** possible reason for these discrepancies **between observed and predicted $\Delta k_h$**…*

*P11L36: Results for these model results should be given, even if just summarized briefly as a % change from steady-state conditions. Also for P12L16. Could a figure for just one site be added to illustrate the difference between steady and non steady state?*

We added the results of the non-steady state simulations on all the soils in the form of a Supplementary figure:

*Surprisingly, the results from this numerical model differed only marginally from those shown in Fig. 5 and Fig. 6 (**see Supplementary Material Fig. S1**).*

*Figure S1: same as Fig. 6 but with $k_{iso}$ values retrieved from the non-steady state model as described in the main text.*

[Figure]

*P12L1: Reader should be pointed to Table 1 to look for phosphate concentrations.*

We added a reference to the table:

*Another factor that could explain the deviation of $\Delta k_h$ from theory is the presence of phosphate ions in the soil solution **(Table 1)** that could either activate or inhibit CA compared to its activity in the absence of such anions (Rowlett et al., 1991; Rusconi et al., 2004).*

*P13L1-16: Interesting results and informative discussion*

Thanks!

*Table 1: citations for literature data should be provided*

References have been added in the Table 1 caption.

*Table 1: main characteristics of the soils investigated in this study. Numbers in italics indicate literature data **(Achat et al. 2014)**.*

*Figure 1: the 'automatic trigger' terminology seems a bit odd if the text just calls the component a 3-way valve*

Figure legend changed, with "3-way valve" instead of "automatic trigger".

*Figure 1: Schematic of the experimental setup used to estimate simultaneously the $CO_2$-$H_2O$ isotope exchange rate ($k_{iso}$) in a soil microcosm and the oxygen isotopic composition of the soil water pool with which the $CO_2$ equilibrates ($\delta_{sw-eq}$). The soil microcosm **consists of 280–300 g of dry soil previously re-humidified to 25% of the water holding capacity using mineral water containing different amounts of exogenous CA powder. The soil column** is thermally regulated using a **6.5L** water bath and **the air entering the chamber is a mixture of** $CO_2$ in dry air whose oxygen isotopic composition **is** alternatively enriched (steady state 1, -3.8‰ VPDBg) and depleted (steady state 2, between -24‰ and -27‰ VPDBg, depending on the experiment).*

[Figure]

Legend:

⌀ mass flow controller (MFC)

⌀ 3-way valve

⌷ manifold

▮ soil

▯ water bath

*Figure 3: What is the basis for the expectation that beta CAs are the most abundant in soils? Provide citation or justification.*

We changed the figure caption slightly and also added a reference:

*These theoretical curves have been obtained using **the un-catalysed rate formula compiled in Uchikawa and Zeebe (2012) and** enzymatic parameters of $k_{cat}/K_M = 70 \ s^{-1} \ \mu M^{-1}$ and $pK_a = 7$, which are typical values **for CA-catalysed $CO_2$ hydration (Rowlett et al. 2002; Smith & Ferry 2000).***

*Figure 4: It is not clear which lines and points in Figure 4 correspond with LeBray1 soil versus an αCA addition of 24 mg L-1, which are both stated in the caption. If the αCA data were plotted for some soils, wouldn't the kiso values be different? If they are not significantly different, as suggested in Fig 6, a justification for plotting results from the no-addition and addition should be given because that reasoning is not clear at the beginning of the results*

*section. Would it be worthwhile switching the order of 3.1 and 3.2 or referencing 3.2 as justification?*

Section 3.1 is required to understand results presented in section 3.2 as it explains, for each soil microcosm and CA treatment, how we were able to retrieve values of $k_{iso}$ and $\delta_{sw\text{-}eq}$. We added a sentence in section 3.1 to reinforce the idea that results shown in Fig. 4 are just an example:

*This approach, when presented graphically, leads to a plot **with** up to six curves (**2 curves per sequence, see Fig. 4 in the case of LeBray1 with 24mg/L of exogenous CA addition**) that intersect at very similar locations within the $k_{iso}$-$\delta_{sw\text{-}eq}$ space.*

We also modified the figure caption:

*Figure 4: The $CO_2$-$H_2O$ isotopic exchange rate ($k_{iso}$) and isotopic composition of soil water equilibrated with $CO_2$ ($\delta_{sw}$) retrieved using the two-steady-state approach described in the main text (**Eqs. 6a and 6b**), for **one single microcosm (LeBray1 with an α-CA addition of 24 mg $L^{-1}$**). Relationships between $k_{iso}$ and $\delta_{sw}$ for steady-state 1 (dotted lines) and steady-state 2 (solid lines) are also shown. In this example **the microcosm was measured** over 3 **consecutive** sequences, resulting in 3 curves for each steady state and 3 intersection points **that coincide well with the two-steady-state solution for each sequence (black squares)**.*

*Figure 5: May be useful to state why plotted without distinction (CA conc shouldn't affect result for water isotopic composition) and restate why $CO_2$ gas exchange results shift with depth (Eq xx)*

Caption of Figure 5 has been amended accordingly:

*Figure 5: The isotopic composition of soil water at different depths in the replicated soil microcosms from each site, estimated either by vacuum distillation and water isotope analysis (**$\delta_{sw}$**, blue squares) or online $CO_2$-$H_2O$ isotopic exchange using the two steady-state approach (**$\delta_{sw\text{-}eq}$, at depth $z_{eq}$**, black circles). Profiles for the different CA treatments are plotted together without distinction (**because exogenous CA addition should not affect the isotopic composition of soil water**). The blue vertical line also indicates the isotopic composition of the irrigation water used for the re-wetting of the air-dried soils. **According to Eq. 11, the addition of exogenous CA shifts the gas exchange results ($\delta_{sw\text{-}eq}$) to shallower depths ($z_{eq}$).***

*Figure 6: difficult to see diamond points - shift CA concentration labels. Why are some LeBray2 points missing in 6b? What are the open circles representing? State in caption.*

Figure has been redrawn with shifted CA concentration labels and the fit to the "native" $k_{iso}$ values has been modified (no extrapolation outside the measured pH range, polynomial fit rather than a spline fit) which led to a smoother "basal" line. The associated caption has also

been changed to:

*Figure 6: **(a)** measured $CO_2$-$H_2O$ isotopic exchange rates ($k_{iso}$) in the different soils for different levels of α-CA addition and **(b)** associated enhancement hydration rates ($k_h$ - $k_{h,native}$) caused by the α-CA addition. In panel a, the un-catalysed isotope exchange rate ($k_{iso,uncat}$, **see Uchikawa and Zeebe (2012))** is shown for reference (black dotted curve). **The pH dependence of the native isotope exchange rates (grey points in panel a) is interpolated over the entire pH range explored here using a third-order polynomial fit (grey curve in panel a). The range of the theoretical rates above this native rate curve that we would expect from αCA addition of 24mg/L (purple curve and hatched area) and 80mg/L (green curve and hatched area) are also shown and have been obtained using $k_{cat}/K_M = 30 \pm 5\ s^{-1}\ \mu M^{-1}$ and $pK_a = 7.1 \pm 0.5$. For those microcosms that were measured multiple times (several sequences), smaller open symbols are displayed to indicate the results from each individual sequence. In some cases, (e.g. LeBray 2), some points could not be displayed in panel b because the $k_{iso}$ measured after CA addition was smaller than the mean native $k_{iso}$, resulting in negative $\Delta k_h$ values (within the measurement uncertainty).***

[Figure]

---

## Author Comment (AC2) · 20 Nov 2017

**Response to Anonymous Referee #3[*]**

[*]Extracts from reviewer's original comments are indicated in *blue italic*

[*]Extracts from our original manuscript are indicated in *black italic*

[*]Proposed modification on our original manuscript are indicated in ***black bold italic***

*The stable isotope of oxygen ($\delta^{18}O$) is routinely used to characterize ecosystem processes from the cell to the globe (Werner et al., 2011, Biogeosciences). The enzymatic activity of carbonic anhydrase (CA) enhances the exchange of the source water $\delta^{18}O$ signal with $CO_2$ emitted from points of respiration. We can better constrain estimates of ecosystem metabolism by increasing our knowledge of CA activity under different environmental conditions. Sauze et al. set out to fill these gaps related to soil in our understanding through a series of innovative experiments and modelling. They quantified the isotopic exchange rate ($k_{iso}$) between $H_2O$ and $CO_2$ in soils with a range of pH. The authors devised small incubation units to which they could add water of known isotopic composition, known $CO_2$ gas, and varying amounts of α-CA (28 and 80 mg $L^{-1}$). By controlling the environment, mainly through maintaining a near constant relative humidity, the authors were able to isolate the impact of soil pH on CA as determined by $k_{iso}$. They found that the experimental results from three of the soils were congruent with their modelling. For the other three soils that did not conform to the modelling prediction, the authors infer that heterogeneity from the soil column, non-steady state conditions, or phosphate ions may have led to the aberrations.*

We are pleased that referee #3 appreciated the originality and significance of our study.

*The experiment was carried out well and the analyses were quite thorough, although a few statistical analyses with regards to differences in the isotopic composition of the source water and extracted water are needed.*

Referee #3 refers here to results shown in Fig. 5 on the isotopic composition of soil water estimated either from soil water vacuum extraction or from $CO_2$ gas exchange. We agree that it would be good to perform statistical analyses to test the significance of the difference between the two estimates. However, the two estimates are also depth-dependent which complicates the exercise, notably when $CO_2$ gas exchange values fall below the deepest point sampled for water extraction (e.g. LeBray1 and LeBray2). It is also not possible to perform such a test on soils where we only had one microcosm per treatment. For the soils that were measured in triplicates for each CA treatment, we tested for significance between the mean ($n = 3$) $\delta_{sw-eq}$ values and the mean ($n = 3$) $\delta_{sw}$ values measured over the entire soil column and weighted by soil moisture content. It turns out that all the soils tested present a significant difference between $\delta_{sw}$ and $\delta_{sw-eq}$. These statistical test results are now gathered into a new table (Table S2).

| Soil name | CA treatment | $\langle\delta_{sw}\rangle$ | $\langle\delta_{sw\text{-}eq}\rangle$ | $n$ | $t$-test |
|---|---|---|---|---|---|
| Le Bray 1 | 0 | -11.97 | -7.31 | 1 | - |
| Le Bray 1 | 24 | -11.24 | -6.88 | 1 | - |
| Le Bray 1 | 80 | -9.57 | -7.59 | 1 | - |
| Le Bray 2 | 0 | -13.56 [a] | -8.79 [b] | 3 | $P < 0.05$ |
| Le Bray 2 | 24 | -13.31 | -8.95 | 3 | $P < 0.05$ |
| Le Bray 2 | 80 | -13.35 | -9.41 | 3 | $P < 0.05$ |
| Planguenoual | 0 | -7.95 | -7.31 | 1 | - |
| Planguenoual | 24 | -5.50 | -7.83 | 1 | - |
| Planguenoual | 80 | -5.97 | -6.40 | 1 | - |
| Pierrelaye | 0 | -11.88 | -9.48 | 3 | $P < 0.05$ |
| Pierrelaye | 24 | -11.40 | -9.73 | 3 | $P < 0.05$ |
| Pierrelaye | 80 | -10.97 | -9.61 | 3 | $P < 0.05$ |
| Folleville | 0 | -8.31 | -7.87 | 1 | - |
| Folleville | 24 | -6.58 | -7.59 | 1 | - |
| Folleville | 80 | -6.95 | -8.11 | 1 | - |
| Toulouse | 0 | -11.03 | -9.68 | 3 | $P < 0.05$ |
| Toulouse | 24 | -10.90 | -9.57 | 3 | $P < 0.05$ |
| Toulouse | 80 | -10.71 | -9.68 | 3 | $P < 0.05$ |

*Table S2: Mean $\delta_{sw}$ measured over the entire soil column and weighted by soil moisture content and corresponding mean $\delta_{sw\text{-}eq}$ for each soil and CA treatment. For LeBray1, Planguenoual and Folleville, one single microcosm was measured over three consecutive gas-exchange sequence, which did not allow us to test for significance differences between the two means. For the other soils, three different microcosms were measured for each treatment, and care was taken to maintain a relatively homogeneous soil water isotopic composition (Fig. 5) so that statistical tests for significant differences could be performed using the open-source software R v.3.3.1 (R Core Team, 2015).*

This table is also referred to in the text at the beginning of section 4.2.

*Our results also revealed large differences between the isotopic composition of the water pool "seen" by the $CO_2$ ($\delta_{sw\text{-}eq}$) and that of cryogenically extracted soil water ($\delta_{sw}$), with significantly ($P < 0.05$) more depleted $\delta_{sw\text{-}eq}$ values compared to $\delta_{sw}$ (Fig. 5 **and Table S2**). Interestingly very similar "offsets" between $\delta_{sw}$ and $\delta_{sw,eq}$ were also predicted by the numerical model (not shown), except for LeBray1 where even larger offsets were found. For a given soil the offset did not seem to vary with soil CA activity (i.e. the difference between $\delta_{sw}$ and $\delta_{sw\text{-}eq}$ was the same for soils with and without CA addition**, see Table S2**) and, at least for the only soil tested, did not seem to be affected by **small changes in** soil water content (similar offsets were observed between LeBray1 and LeBray2). However, in-between the different soils, it seemed that those with the highest CA activity (Planguenoual, Folleville) also had the smallest offset **(Table S2)**. **Also, for LeBray soil, Jones et al. (2017) showed that the offset between $\delta_{sw}$ and $\delta_{sw,eq}$ decreased when the soil was approaching saturation.**

*The manuscript needs a better description of the experiments performed and the hypotheses/questions asked of each steady-state iteration. I suggest a simple table or figure will suffice. This will let the reader keep better track of 1) some of the terminology used to describe different parameters (e.g., native, non-native), 2) sample size, and the results in general.*

The distinction between native and non-native is now clearly defined in the text (see comment below related to page 4 line 30). We also modified the caption of Figure 1 to re-precise sample size:

*Figure 1: Schematic of the experimental setup used to estimate simultaneously the $CO_2$-$H_2O$ isotope exchange rate ($k_{iso}$) in a soil microcosm and the oxygen isotopic composition of the soil water pool with which the $CO_2$ equilibrates ($\delta_{sw-eq}$). The soil microcosm* **consists of 280–300 g of dry soil previously re-humidified to 25% of the water holding capacity using mineral water containing different amounts of exogenous CA powder. The soil column** *is thermally regulated using a* **6.5L** *water bath and* **the air entering the chamber is a mixture of** *$CO_2$ in dry air whose oxygen isotopic composition* **is alternatively** *enriched* **(steady state 1, -3.8‰ VPDBg) and** *depleted* **(steady state 2**, *between -24‰ and -27‰ VPDBg, depending on the experiment).*

*In the introduction the real-world differences between α- and β-CA should be discussed. Furthermore, the use of α-CA can be defended more strongly upfront rather than toward the conclusion of the manuscript.*

We added a couple of sentences in the introduction to discuss differences between α- and β-CA:

*Changes in the abundance and diversity of soil microbial communities were proposed as possible drivers of the observed spatial and temporal changes in soil CA activity (Seibt et al., 2006; Wingate et al., 2008, 2009, 2010). In particular, soil pH is known to strongly influence microbial community composition, richness and diversity (Fierer and Jackson, 2006; Griffiths et al., 2011; Hartman et al., 2008; Lauber et al., 2009) and could thus influence soil CA activity indirectly via changes in the microbial populations* **that would translate into differences in CA requirements and in the expression of classes of CA with different enzymatic efficiencies**. **Indeed, α- and β-CA classes are not represented equally in all kingdoms. Very schematically, α-CAs tend to be more abundant in algae and micro-algae while β-CAs are more commonly found in fungi (Elleuche & Poggler 2010; Moroney et al., 2001). In addition, α-CAs can be extracellular enzymes unlike β-CAs that are, to our knowledge, only intracellular enzymes.**

The choice of using α-CA for our CA addition treatment is also defended more strongly in the introduction (page 3 line 35):

"*This CA isoform was chosen because it is well characterised in terms of enzymatic activity (Uchikawa and Zeebe, 2012) and pH response (Rowlett et al., 1991*) **and it has been demonstrated that its activity was very stable in time even after several hours in solution (Uchikawa and Zeebe 2012)**.*"

*I also am surprised by the differences in $\delta^{18}O$ between the soil extracted and equilibrated water values. This is important evidence in the on-going debate of the two water world hypotheses (Kirchner, James W. "A double paradox in catchment hydrology and geochemistry." Hydrological Processes 17, no. 4 (2003): 871-874.; McDonnell, J. J. (2014), The two water worlds hypothesis: ecohydrological separation of water between streams and trees?. WIREs Water, 1: 323–329.; Sprenger, M., H. Leistert, K. Gimbel, and M. Weiler (2016), Illuminating hydrological processes at the soil-vegetation-atmosphere interface with water stable isotopes, Rev. Geophys., 54, 674– 704.Vargas, A. I., Schaffer, B., Yuhong, L. and Sternberg, L. d. S. L. (2017), Testing plant use of mobile vs immobile soil water sources using stable isotope experiments. New Phytologist, 215: 582–594.).*

Initially, we did not want to put too much emphasis on these reported $\delta^{18}O$ differences because our study was mostly focusing on the pH response of the $CO_2$-$H_2O$ isotopic exchange rate $k_{iso}$. We agree however that this "side" result on $\delta^{18}O$ is also interesting and we devoted in the end half of the discussion to this finding. In response to a previous comment we have now included a new table (Table S2) reporting results for significant differences between the two $\delta^{18}O$ estimates. We believe it is enough "publicity" for this finding. We decided not to refer to the on-going debate on the "two water world hypothesis" because this would bring too much diversion and also raise some unanswered questions (does gaseous $CO_2$ "visit" the same soil water pool as the one tapped by plant roots during water uptake?).

*Page 1 Line 12: delete "nonetheless"*

It is important to tell the reader this lack of understanding is currently a weakness of the $^{18}O$ methodology to partition gross $CO_2$ fluxes.

*Page 2 Line 3: Biosphere "absorbs" - unconventional way of describing flux and pools*

We replaced "absorbs" by "mitigates":

*The terrestrial biosphere currently **mitigates** about 25% of anthropogenic $CO_2$ emissions as a result of a small disequilibrium **between two large gross $CO_2$ fluxes, photosynthetic $CO_2$ uptake and respiratory $CO_2$ release** (Le Quéré et al., 2015).*

*Page 2 Line 17: delete "advanced over recent years" given the literature list dating back to 1997 (and earlier?), the method is established.*

Done.

*Page 2 Line 37: perhaps simple instead of crude.*

Done.

*Page 2 Line 40: since you introduce these values can you report the accepted global photosynthesis rates?*

This is now done:

*Subsequently, using the lower range of soil CA activity estimates made by Wingate et al. (2009), an atmospheric $CO^{18}O$ inversion was performed and led to a rate of global photosynthesis of ca. 175 GtC $yr^{-1}$ over the period 1980-2010 (Welp et al., 2011), a* **surprisingly high value compared to the accepted global estimate of 115-130 GtC $yr^{-1}$ (Beer et al., 2010; IPCC 2013)**.

*Page 3 Line 4: You introduce microbial communities as a possible explanation, but not how the microbes might alter CA activity. I assume it might have something to do with total biomass and functional characteristics of the communities.*

We have clarified this point now by adding two extra sentences referring to the different classes of CA, their abundance in different kingdoms and their differences in intrinsic activities (i.e. enzymatic parameters). See general comment above about the introduction.

*Page 3 Line 18: please provide a citation for this value.*

We added a reference to the text:

*This is because soil CA activities are often reported relative to the un-catalysed $CO_2$-$H_2O$ isotopic exchange rate ($k_{iso,uncat}$), usually assumed equal to ca. 0.012 $s^{-1}$ at 25°C* **(Miller et al., 1999)**.

*Page 3 Line 23: I think you mean soil solution here. Perhaps specify in your hypotheses that you are primarily interested in the direct effects of pH and not the indirect effects anticipated from shifts in microbial diversity and function.*

Indeed we meant to refer to the chemical composition of the soil "solution". We also modified the hypothesis to specify that we primarily looked at soil pH because of its (direct) role on $CO_2$ speciation:

*The chemical composition of the soil* **solution** *is another potentially important factor that should be considered when reporting soil CA activity. [...]*
**Because of the direct role of pH on $CO_2$ speciation and $CO_2$ hydration rate**, *we hypothesised that exogenous CA activity should be inhibited in acidic soils, but that the native soil phosphate concentration might also influence the activity of CA for soils differing in pH.*

*Page 4 Line 30: I suggest to introduce the "native" or control term here and please explain the situation to which you arrived at estimates of "un-catalysed rates" (page 9 line 28).*

We now clarified in the text what the terms "native" and "non-native" mean:

**CA activities from soil microcosms without CA powder addition are qualified hereafter as "native" and CA activities related to the CA addition are called "non-native" and estimated, for a given soil, as the difference between the activities on the CA-added microcosms and their native rates.**
We theoretically estimated the uncatalysed rate depending on temperature and pH and simply

compared our measured isotopic exchange rates compared to the theory. A reference to the theoretical calculation (Uchikawa and Zeebe 2012) is now given.

*Page 4 Line 32: Why were these concentrations chosen?*

We chose these CA concentrations because they should correspond to the upper range of CA concentrations in natural soils (Ogée et al. 2016). This is now explained in the text:

*For a set of gas exchange measurements, lyophilised α-CA powder from bovine erythrocytes (C3934-100MG, Sigma-Aldrich, France) was diluted into the irrigation water. For each set of experiments, CA concentrations of ca. 24 and 80 mg $L^{-1}$ were used.* **We chose these concentrations because they would correspond to the upper range of CA concentrations expected in natural soils, assuming a cytoplasmic CA concentration of 0.1mM (Ogée et al. 2016)**.

*Page 4 Lines32-33: Please edit this sentence, especially "the measurement soils without".*

We modified the sentence as follows:

*Apart from this addition of CA into the irrigation water, all other* **preparation** *steps* **of the soil microcosms** *were kept identical* **to the ones** *described above for* **the microcosms** *without CA addition.*

*Page 4 Line 38: Introduce the water bath here and how its temperature was maintained.*

This is now done:

*Prior to gas exchange measurements, each soil pot was* **closed using a screw-tight lid connected to inlet and outlet tubes (Fig. 1) and immerged into a 6.5L water bath, thermally-regulated at 20°C**. *An acclimation time of at least 20 minutes was used to allow the soil column to re-equilibrate to the new air supply $CO_2$ composition* **and the new temperature**. *The soil $CO_2$ efflux and its oxygen isotopic composition were then measured using the experimental setup illustrated in Fig. 1.*

*Page 5 Line 3: The description of the two gases here is lacking. Is there not also $CO_2$ in the compressed air-tank? It is not clear at this point why you have the two different gas sources. Please explain how you achieved different $\delta^{18}O$ compositions in the two gas sources.*

We now explain in more details why we used different $\delta^{18}O$ composition for the two air supplies corresponding to the two steady states (SS1 and SS2) and how we did it:

*To simultaneously retrieve soil CA activity, reported here as the $CO_2$-$H_2O$ isotopic exchange rate $k_{iso}$, and the $\delta^{18}O$ of the soil water pools with which $CO_2$ equilibrates ($\delta_{sw-eq}$), we designed a system that allowed us to measure $CO_2$ isotope fluxes under two, quasi-simultaneous isotopic steady states that only differ in the isotopic composition of the $CO_2$ entering the soil chamber (Fig. 1). The air supplied to the chamber came directly from a* **tank containing dry** *air during the*

*first steady state (SS1) and from a mix of dry, CO₂-free air and a tank of pure CO₂ during the second steady state (SS2).* **In practice, the air was supplied to the microcosms during SS2 using a compressor (FM2 Atlas Copto, Nacka, Sweden), coupled to a chemical scrub column (Ecodry K-MT6, Parker Hannifin, Cleveland, OH, US) that removed water vapour and CO₂ from the air before being mixed with pure CO₂, with a $\delta^{18}O$ isotopic compositions significantly different from the CO₂-in-air mixture used in SS1.** *During SS2 mixing valves adjusted the CO₂ concentration of the inlet air to maintain it close to the value of the inlet air used in SS1 within acceptable error (423 ± 5 ppm), whilst their oxygen isotope compositions differed markedly (Fig. 2). The transition between SS1 and SS2 was operated by means of a three-way valve (Fig. 1) and a transition period of 20 minutes was necessary to attain the new steady state (Fig. 2).*

*Page 5 Line 24: each is singular in this case: "Each line was measured", "only the last 40s of measurement was averaged"*

Corrected:

*To minimise carry-over effects caused by this residence time, each line (inlet or chamber air or calibration tanks)* **was** *measured for 2 minutes and only the last 40 s of measurements* **were** *averaged to provide a single mean and standard deviation.*

*Page 5 Line 29: do you mean over the measurement time? i.e., the measurement period?*

The measurement and mean and standard deviation calculations on the calibration tank are the same as for the inlet or chamber air *i.e.* we measured them 2 minutes and used only the last 40 s for our calculations. This is now clarified in the text:

*Calibration tank mixing ratios for the different isotopologues ($^{12}CO_2$, $^{13}CO_2$ and $CO^{18}O$) were averaged as described above* **(2 minutes of measurement and only the last 40 s were averaged)** *and interpolated in time using a spline function.*

*Page 6 Line 1: what do you mean precisely by "were propagated"?*

We meant "mathematically propagated". We changed the wording, with the hope that the way we reported measurement errors is now clearer:

*Standard deviations on CO₂ mixing ratios of the different isotopologues were* **used to compute measurement error on total CO₂ concentration***.*

*Page 6 Line 17: The equation in parentheses is difficult to decipher, perhaps separate it from the text.*

We could include this equation into Eq. 4 but we are afraid that it would make the new equation too long and difficult to read. Because the equation in the text is just a definition while the equation we used to compute $\delta_F$ is what is written in Eq. 4 we felt more important to emphasize Eq. 4. We thus preferred to keep things as they are.

*Page 6 Line 25: Can you better define piston velocity here? It looks like eqn. 5 is a formulation of Fick's diffusion implementing Henry's law. In this case, is the piston velocity expressing a minimal exchange of gas at the soil-water boundary during equilibrium? Tans (1998) also discusses piston velocity within this context.*

We added a few word to describe $V_{inv}$, using the exact wording used by Tans (1998):

*F is the soil $CO_2$ efflux (µmol $m^{-2}$ $s^{-1}$) and $V_{inv}$ (m $s^{-1}$) is the piston velocity **(i.e. the rate at which a column of air gets pushed into the soil; Tans, 1998)**.*

*Page 6 Line 29: delete "so-called"*

Done.

*Page 7 Line 11: $D_{iso}$ is not defined here*

We now define $D_{iso}$ sooner, i.e. just after Eq. 7:

*...where $z_{max}$ is soil depth and $z_1 = D_{iso}/V_{inv}$ with **$D_{iso} = D_{eff}/(1 - \varepsilon_D)$ and $D_{eff}$ ($m^2$ $s^{-1}$) is the effective diffusivity of gaseous $CO_2$ through the soil matrix (Tans, 1998; Wingate et al., 2010). The latter was computed using the formulation of Moldrup et al. (2003) for repacked soils: $D_{eff} = (\varphi - \theta)^{2.5}/\varphi\ D_0$, where $\varphi$ ($m^3$ $m^{-3}$) is total soil porosity and $D_0$ ($m^2$ $s^{-1}$) is the molecular diffusivity of $CO_2$ in soil air at temperature $T_s$ (K): $D_0 = 1.381\ 10^{-5}\ (T_s/273.15)^{1.81}$ (Massman, 1998).** The right-hand side of Eq. 6b was then used to estimate (...).*
*The soil $CO_2$–$H_2O$ isotopic exchange rate ($k_{iso}$, in $s^{-1}$) was then derived from the piston velocity according to:*

$$k_{iso} = \frac{V_{inv}^2}{D_{iso} B \theta} \qquad\qquad (8)$$

*where B ($m^3$ $m^{-3}$) is the solubility coefficient for $CO_2$ in water (Weiss, 1974) **and** $\theta$ ($m^3$ $m^{-3}$) is the volumetric soil water content.*

*Page 7 Line 28: add carbonyl sulfide along with OCS*

Done.

*Page 8 Line 10: refrigerator instead of "fridge"*

Changed.

*Page 8 Line 25: please edit" "to help vaporise the water under vacuum immediately upon injection", this does not read well.*

We edited the sentence as follows:

*A small water volume (0.2-1.0 µL) from each vial was sampled using a 5-µL syringe (SGE Analytical Science, Ringwood, Australia) and injected through a septum in a vaporiser unit*

*maintained at 80°C to **ensure complete vaporisation of the liquid water straight after injection**.*

*Page 8 Line 28: I assume this is a data filter and not a physical filter used in the analysis.*

Effectively it is a data filter, we edited the sentence:

*Each vial was then measured eight times in total and only the last five measurements, subject to **data** filtering, were retained and averaged.*

*Page 8 Line 29: "of the measurements"; please edit this whole sentence.*

We edited the sentence:

***Based on measurements on the internal standard used for quality check,** the accuracy (**i.e.** the mean absolute difference between calibrated and true $\delta^{18}O$ values) and reproducibility (**i.e.** the standard deviation of these means) **of** our $\delta^{18}O$ measurements were always below 0.15 ‰ and 0.1 ‰ respectively.*

We changed this sentence (also to address reviewer 2's comment) and replaced near-common to very similar:

*This approach, when presented graphically, leads to a plot **with** up to six curves (**2 curves per sequence, see Fig. 4 in the case of LeBray1 with 24mg/L of exogenous CA addition**) that intersect at **very similar locations** within the $k_{iso}$-$\delta_{sw\text{-}eq}$ space.*

*Page 9 Line 8: perhaps, inform the reader before the calculation is described that it is desirable to verify the $k_{iso}$ and $\delta_{sw\text{-}eq}$ independently.*

We believe that this comment refers to the following sentence where we talk about estimating $k_{iso}$ and $\delta_{sw\text{-}eq}$ "separately" (as opposed to "graphically"). We realised that the wording was misleading and replaced it by "numerically":

*Combining the two steady states from the same sequence and using the iterative procedure described above, it is also possible to estimate $k_{iso}$ and $\delta_{sw\text{-}eq}$ **numerically**, as **indicated** by the symbols in Fig. 4.*

The word "sequence" refers here to a full sequence of measurement as described in Fig.2. This is now specified in the text (the notion of "sequence" is also better explained earlier in the text, in response to some comments from reviewer 2):

*Combining the two steady states from the same sequence **of measurement (Fig.2)** and using the iterative procedure described above, it is also possible to estimate $k_{iso}$ and $\delta_{sw-eq}$ separately, as demonstrated by the symbols in Fig. 4.*

*Page 9 Line 20: the difference is roughly 1 per mil, can you check that the difference is significant?*

Our estimations of $\delta_{sw-eq}$ were always significantly different from the $\delta^{18}O$ value of the irrigation water and the cryogenically-extracted waters (t-test, $P < 0.05$). The results from these statistical tests have been incorporated into the revised manuscript (a table has also been added in the Supplementary material, see above):

*These estimated values of $\delta_{sw-eq}$ were **significantly different (P < 0.05) from** the $\delta^{18}O$ of irrigation water (-10.1 ‰VSMOW) **and from the mean, cryogenically-extracted soil water averaged over the entire soil column and weighted by volumetric soil water content** (Fig. 5).*

*Page 9 Line 28: why native? Is this the same as the control?*

The term "native" is associated to the soil microcosms where no CA was added (it could also have been called control). This was clarified in section 2.2 and is re-explained again at the very beginning of section 3.2 (see above).

The un-catalysed rate and the referring symbol $k_{iso,uncat}$ is introduced in the introduction (page 3) but we felt the need to recall the symbol and its meaning here to help the reader understand Fig. 6.

*Page 10 Equation 12: let the reader know that $(k_{cat}/K_M)_{max}$ is explained previously in equation 10.*

This is now done:

*This influence of soil pH on the enhancement of $k_h$ by exogenous CA was anticipated as the $k_{cat}/K_M$ **(appearing in Eq. 10)** is known to be strongly reduced in acidic pH with a pH response of the form (Rowlett et al., 1991):*

*Page 10 Line 33: I suggest to separate Fig. 6a and 6b. Is the apparent peak and subsequent decline of $k_{iso}$ explained in the results or the discussion?*

We are convinced that Figs. 6a and 6b should stay together as we derived the latter from the former. The optimum in $k_{iso}$ with pH is clearly explained in the text referring to the theoretical curves drawn in Fig. 3 (end of section 2.5 and beginning of section 3.2).

*Page 10 Line 19: This is a little confusing, it can be read as though you are using exogenous CA as a tool to predict the enhancement in soil CA activity, or it can be read as a general question asking if we can predict the enhancement in soil CA activity when additional CA is introduced.*

We agree that the title of the subsection was confusing. We changed it to:

*Can we predict the enhancement in soil CA activity **associated with** exogenous CA **addition**?*

*Page 10 Line 20: Since this is the first sentence of your discussion, perhaps you can set the reader up for what the topic of discussion is for this paragraph. The term native doesn't appear until page 9 line 28 and I don't think the term non-native is ever defined in the prior text. This exemplifies why a table or figure explaining the experiment will help the reader.*

We now define more clearly what we mean by native and non-native rates in the Material and Methods, and the definition of the non-native rate is also re-established in this first sentence of the discussion.

*Page 10 Line 28: please explain "native hydration". Is this potential water remaining within the soil from the field or elsewhere that has potentially mixed with the irrigation water?*

See comment above.

*Page 11 Line 11: this sentence is a little convoluted, it reads at first as if the soil pH is going to have a response when in fact this metric is intrinsic to the soil.*

We do not understand this comment. There is no reference to pH in this sentence or any other sentence before and after this passage.

*Page 11 Line 22: deviations from non-steady state instead of non-steadiness.*

Changed.

*Page 11 Line 26: "was run" or simply "ran"*

Corrected.

*The check for non-steady state was cursory and not all the assumptions were easily understood. While the exercise is interesting, it is not possible for the reader to determine how robust the results are. I think the authors need to decide how important this issue is to their results and either fully address the issue to the best extent possible or shorten the explanation and report that the non-steady state effect needs to be addressed by further experimentation and modelling.*

The results to the non-steady-state simulations are now provided in the Supplementary material.

*Figure S1: same as Fig. 6 but with kiso values retrieved from the non-steady state model as described in the main text.*

[Figure]

*Page 13 - Conclusion: Can you bring out the larger relevance once more? How might the effect of pH influence our estimates of ecosystem carbon balance at different scales?*

We added an extra sentence on the relevance of this work in a broader context:

*Our experimental results demonstrate that our two steady-state approach is robust and sensitive enough to detect changes in the $CO_2$-$H_2O$ isotope exchange rate when the concentration of CA enzyme in the soil matrix is augmented artificially. We also found that natural variations in soil pH had a strong control over the variability of soil CA activity, with a smaller influence of the phosphate ion concentration, and these variations reassuringly followed similar patterns to those observed in other studies on α-CA activity in buffered solutions.* **This is a real advancement in our understanding of the spatial variations of soil CA activity across biomes reported by**

***Wingate et al. (2009) and the associated impact on the atmospheric budget of CO18O. However, our results should still be taken with caution***. *Although α-CAs may be present in certain soil microbial communities with a high abundance of phototrophs such as cyanobacteria and micro-algae, the majority of microbial CAs in soils are more likely represented by the β-CA class (Smith and Ferry, 2000).*

*Figure 2. Please insert the step number within the top or bottom panel. Check the grammar within the caption; for example, I believe you want to say you measured the two calibration bottles in step 1.*

Done:

*Figure 2: Typical time-series of the measured $CO_2$ mixing ratio and isotope composition ($\delta^{18}O$) over the course of a working sequence. The sequence is composed of 7 steps **(indicated in panel b) to successively measure**: (1) two calibration bottles spanning the expected range of $CO_2$ mixing ratios, (2) inlet and outlet lines of the soil microcosm, measured 4 times consecutively, using a $CO_2$ with an enriched $\delta^{18}O$ (steady state 1), (3) calibration bottles, (4) the outlet of the chamber during the switch of the air supplying the soil chamber (front), (5) calibration bottles, (6) inlet and outlet lines of the soil chamber, measured 4 times consecutively, using a $CO_2$ with a depleted $\delta^{18}O$ (steady state 2) and (7) calibration bottles.*

[Figure]

*Figure 3. I recommend that the use of a-CA in the experiments is clarified in the figure. Perhaps also how this figure is related to the experimental results.*

We added a sentence in the caption to say that using parameter values more typical of $\alpha$–CA would not change qualitatively the figure. We also gave the reference for the uncatalysed rate calculations:

*Figure 3: Theoretical rates of $CO_2$ hydration ($k_h$) and $CO_2$-$H_2O$ oxygen isotope exchange ($k_{iso}$) as a function of pH, for 3 levels of carbonic anhydrase concentration. These theoretical curves have been obtained using **the un-catalysed rate formula compiled in Uchikawa and Zeebe (2012) and** enzymatic parameters of $k_{cat}/K_M = 70\ s^{-1}\ \mu M^{-1}$ and $pK_a = 7$, which are typical values **for CA-catalysed $CO_2$ hydration (Rowlett et al. 2002; Smith & Ferry 2000). Using enzymatic parameter values more specific to the $\alpha$CA powder used here for the CA treatment (i.e. $k_{cat}/K_M = 30 \pm 5\ s^{-1}\ \mu M^{-1}$ and $pK_a = 7.1 \pm 0.5$), would not change qualitatively this figure.***

*Figure 4. It is probably worthwhile to report the results for all the soils or at least place them in the supplementary. Please reference where the $k_{uncat}$ is presented in the text.*

The $\delta_{sw}$ values are now provided in a supplementary table and a reference for the $k_{uncat}$ calculations has been added to the figure caption:

*Figure 4: The $CO_2$-$H_2O$ isotopic exchange rate ($k_{iso}$) and isotopic composition of soil water equilibrated with $CO_2$ ($\delta_{sw}$) retrieved using the two-steady-state approach described in the main text, for LeBray1 soil and an $\alpha$CA addition of 24 mg $L^{-1}$. Relationships between $k_{iso}$ and $\delta_{sw}$ for steady-state 1 (dotted lines) and steady-state 2 (solid lines) are also shown. In this example 3 sequences were used, resulting in 3 curves for each steady state and 3 intersection points. The pH-dependent, un-catalysed $CO_2$-$H_2O$ isotopic exchange rate **(Uchikawa and Zeebe 2012)** is also indicated by the grey horizontal line.*

*Figure 6. The caption does not reference a or b. In the text, I could not find a reference to a. Please explain what appears to be model uncertainty. How is the fact that three of your soils did not conform to your model reconciled within fig 6?*

We clarified this figure caption that clearly lacked a lot of information:

*Figure 6: (a) measured $CO_2$-$H_2O$ isotopic exchange rates ($k_{iso}$) in the different soils for different levels of α-CA addition and (b) associated enhancement hydration rates ($k_h$ - $k_{h,native}$) caused by the α-CA addition. In panel a, the un-catalysed isotope exchange rate ($k_{iso,uncat}$ see Uchikawa and Zeebe (2012)) is shown for reference (black dotted curve). **The pH dependence of the native isotope exchange rates (grey points in panel a) is interpolated over the entire pH range explored here using a third-order polynomial fit (grey curve in panel a). The range of the theoretical rates above this native rate curve that we would expect from $\alpha$−CA addition of 24mg/L (purple curve and hatched area) and 80mg/L (green curve and hatched area) are also shown and have been obtained using $k_{cat}/K_M = 30 \pm 5$ $s^{-1}$ $\mu M^{1}$ and $pK_a = 7.1 \pm 0.5$. For those microcosms that were measured multiple times (several sequences), smaller open symbols are displayed to indicate the results from each individual sequence. In some cases, (e.g. LeBray 2), some points could not be displayed in panel b because the $k_{iso}$ measured after CA addition was smaller than the mean native $k_{iso}$, resulting in negative $\Delta k_h$ values (within the measurement uncertainty).***

A reference to panel 6a is now done in the text (page 9, lines 30-33):

*The addition of exogenous CA generally led to higher $k_{iso}$ values compared to the native rates, and also enhanced $CO_2$ hydration rates $k_h$, with marked differences depending on the pH range (Fig. 6a).*

The discrepancies between some data points shown in panel b and the theory is clearly addressed in the discussion of the paper.